# Multi-task Batch Reinforcement Learning with Metric Learning

**Jiachen Li**[1][*]  **Quan Vuong**[1][*]  **Shuang Liu**[1]  **Minghua Liu**[1]
**Kamil Ciosek**[2]  **Henrik Christensen**[1]  **Hao Su**[1]
[1]UC San Diego     [2] Microsoft Research Cambridge, UK
{jil021, qvuong, s3liu, minghua, hichristensen, haosu}@ucsd.edu
kamil.ciosek@microsoft.com

## Abstract

We tackle the Multi-task Batch Reinforcement Learning problem. Given multiple datasets collected from different tasks, we train a multi-task policy to perform well in unseen tasks sampled from the same distribution. The task identities of the unseen tasks are not provided. To perform well, the policy must infer the task identity from collected transitions by modelling its dependency on states, actions and rewards. Because the different datasets may have state-action distributions with large divergence, the task inference module can learn to ignore the rewards and spuriously correlate *only* state-action pairs to the task identity, leading to poor test time performance. To robustify task inference, we propose a novel application of the triplet loss. To mine hard negative examples, we relabel the transitions from the training tasks by approximating their reward functions. When we allow further training on the unseen tasks, using the trained policy as an initialization leads to significantly faster convergence compared to randomly initialized policies (up to $80\%$ improvement and across 5 different Mujoco task distributions). We name our method **MBML** (**M**ulti-task **B**atch RL with **M**etric **L**earning) [2].

## 1 Introduction

Combining neural networks (NN) with reinforcement learning (RL) has led to many recent advances [1–5]. Since training NNs requires diverse datasets and collecting real world data is expensive, most RL successes are limited to scenarios where the data can be cheaply generated in a simulation. On the other hand, offline data is essentially free for many applications and RL methods should use it whenever possible. This is especially true because practical deployments of RL are bottle-necked by its poor sample efficiency. This insight has motivated a flurry of recent works in Batch RL [6–10]. These works introduce specialized algorithms to stabilize training from offline datasets. However, offline datasets are not necessarily diverse. In this work, we investigate how the properties of a diverse dataset influence the policy search procedure. By collecting diverse offline dataset, we hope the networks will generalize without further training to unseen tasks or provide good initialization that speeds up convergence when we perform further on-policy training.

To collect diverse datasets, it occurs to us that we should collect data from different tasks. However, datasets collected from different tasks may have state-action distributions with large divergence. Such dataset bias presents a unique challenge in robust task inference. We provide a brief description of the problem setting, the challenge and our contributions below. For ease of exposition, we refer to such datasets as having little overlap in their state-action visitation frequencies thereafter.

---

[*]Equal Contribution

[2]Website: https://sites.google.com/eng.ucsd.edu/multi-task-batch-reinforcement/home

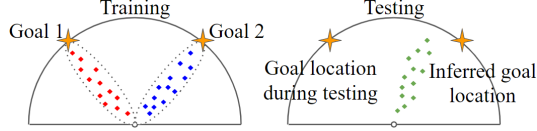

Figure 1: A toy example to illustrate the challenge. The agent must navigate from the origin to a goal location. **Left:** Goal 1 and Goal 2 denote the two training tasks. The red and blue squares indicate the transitions collected from task 1 and 2 respectively. We can train the task inference module to infer the task identity to be 1 when the context set contains the red transitions and 2 when the context set contains the blue transitions. Since there are no overlap between the red and blue squares, the task inference module learns to correlate the state-action pairs to the task identity. **Right:** The failure of the task inference module. The policy must infer the task identity from the randomly collected transitions, denoted by the green squares. The agent needs to navigate to goal 1 during testing. However, if the green squares have more overlap with the blue squares, the task inference module will predict 2 to be the task identity. The agent therefore navigates to the wrong goal location.

We tackle the Multi-task Batch RL problem. We train a policy from multiple datasets, each generated by interaction with a different task. We measure the performance of the trained policy on unseen tasks sampled from the same task distributions as the training tasks. To perform well, the policy must first infer the identity of the unseen tasks from collected transitions and then take the appropriate actions to maximize returns. To train the policy to infer the task identity, we can train it to distinguish between the different training tasks when given transitions from the tasks as input. These transitions are referred to as the context set [11]. Ideally, the policy should model the dependency of the task identity on both the rewards and the state-action pairs in the context set. To achieve this, we can train a task identification network that maps the collected experiences, including both state-action pairs and rewards, to the task identity or some task embedding. This approach, however, tends to fail in practice. Since the training context sets do not overlap significantly in state-action visitation frequencies, it is possible that the learning procedure would minimize the loss function for task identification by *only* correlating the state-action pairs and ignoring rewards, which would cause mistakes in identifying testing tasks. This is an instance of the well-known phenomena of ML algorithms cheating when given the chance [12] and is further illustrated in Fig. 1. We limit our explanations to the cases where the tasks differ in reward functions. Extending our approach to task distribution with different transition functions is easily done. We provide experimental results for both cases.

Our contributions are as follows. To the best of our knowledge, we are the first to highlight the issue of the task inference module learning the wrong correlation from biased dataset. We propose a novel application of the triplet loss to robustify task inference. To mine hard negative examples, we approximate the reward function of each task and relabel the rewards in the transitions from the other tasks. When we train the policy to differentiate between the original and relabelled transitions, we force it to consider the rewards since their state-action pairs are the same. Training with the triplet loss generalizes better to unseen tasks compared to alternatives. When we allow further training on the unseen tasks, using the policy trained from the offline datasets as initialization significantly increase convergence speed (up to $80\%$ improvement in sample efficiency).

To the best of our knowledge, the most relevant related work is [6], which is solving a different problem from ours. They assume access to the ground truth task identity and reward function of the testing task. Our policy does not know the testing task's identity and must infer it through collected trajectories. We also do not have access to the reward function of the testing tasks.

## 2   Preliminaries and Problem Statement

**To help the reader follow our explanation, we include a symbol definition table in Appendix A.**

We model a task as a Markov Decision Process $M = (\mathcal{S}, \mathcal{A}, T, T_0, R, H)$, with state space $\mathcal{S}$, action space $\mathcal{A}$, transition function $T$, initial state distribution $T_0$, reward function $R$, and horizon $H$. At each discrete timestep $t$, the agent is in a state $s_t$, picks an action $a_t$, arrives at $s'_t \sim T(\cdot|s_t, a_t)$, and receives a reward $R(s_t, a_t, s'_t)$. The performance measure of policy $\pi$ is the expected sum of rewards $J_M(\pi) = \mathbb{E}_{\tau_M \sim \pi}[\sum_{t=0}^{H-1} R(s_t, a_t, s'_t)]$, where $\tau_M = (s_0, a_0, r_0, s_1, a_1, r_1, \ldots)$ is a trajectory generated by using $\pi$ to interact with $M$.

## 2.1 Batch Reinforcement Learning

A Batch RL algorithm solves the task using an existing batch of $N$ transitions $\mathcal{B} = \{(s_t, a_t, r_t, s'_t) | t = 1, \ldots, N\}$. A recent advance in this area is Batch Constrained Q-Learning (BCQ) [9]. Here, we explain how BCQ selects actions. Given a state $s$, a generator $G$ outputs multiple candidate actions $\{a_m\}_m$. A perturbation model $\xi$ takes as input the state-candidate action and generates small correction $\xi(s, a_m)$. The corrected action with the highest estimated $Q$ value is selected as $\pi(s)$:

$$\pi(s) = \underset{a_m + \xi(s, a_m)}{\arg \max} Q(s, a_m + \xi(s, a_m)), \qquad \{a_m = G(s, \nu_m)\}_m, \qquad \nu_m \sim \mathcal{N}(0, 1). \quad (1)$$

To help the reader follow our discussion, we illustrate graphically how BCQ selects action in Appendix B. In our paper, we use BCQ as a routine. The take-away is that BCQ takes as input a batch of transitions $\mathcal{B} = \{(s_t, a_t, r_t, s'_t) | t = 1, \ldots, N\}$ and outputs three learned functions $Q, G, \xi$.

## 2.2 Multi-task Batch Reinforcement Learning

Given $K$ batches, each containing $N$ transition tuples from one task, $\mathcal{B}_i = \{(s_{i,t}, a_{i,t}, r_{i,t}, s'_{i,t}) | i = 1, \ldots, K, t = 1, \ldots, N\}$, we define the Multi-task Batch RL problem as:

$$\underset{\theta}{\arg \max} \ J(\theta) = \mathbb{E}_{M_i \sim p(M)} \left[ J_{M_i}(\pi_\theta) \right], \quad (2)$$

where an algorithm only has access to the $K$ batches and $J_{M_i}(\pi)$ is the performance of the policy $\pi$ in task $i$, i.e. $\mathbb{E}_{\tau_{M_i} \sim \pi}[\sum_{t=0}^{H-1} R(s_{i,t}, a_{i,t}, s'_{i,t})]$. $p(M)$ defines a task distribution. The subscript $i$ indexes the different tasks. The tasks have the same state and action space and only differ in the transition and reward functions [13]. A distribution over the transition and/or the reward functions therefore defines the task distribution. We measure performance by computing average returns over unseen tasks sampled from the same task distribution. The policy is not given identity of the unseen tasks before evaluation and must infer it from collected transitions.

In multi-task RL, we can use a task inference module $q_\phi$ to infer the task identity from a context set. The context set for a task $i$ consists of transitions from task $i$ and is denoted $\mathbf{c}_i$. The task inference module $q_\phi$ takes $\mathbf{c}_i$ as input and outputs a posterior over the task identity. We sample a task identity $\mathbf{z}_i$ from the posterior and inputs it to the policy in addition to the state, i.e. $\pi(s, \mathbf{z}_i)$. We model $q_\phi$ with the probabilistic and permutation-invariant architecture from [11]. $q_\phi$ outputs the parameters of a diagonal Gaussian. For conciseness, we sometimes use the term policy to also refer to the task inference module. It should be clear from the context whether we are referring to $q_\phi$ or $\pi$.

We evaluate a policy on unseen tasks in two different scenarios: (1) Allowing the policy to collect a small number of interactions to infer $z$, we evaluate returns without further training, (2) Training the policy in the unseen task and collecting as much data as needed, we evaluate the amount of transitions the policy needs to collect to converge to the optimal performance.

We assume that each batch $\mathcal{B}_i$ contains data generated by a policy while learning to solve task $M_i$. Thus, if solving each task involve visiting different subspace of the state space, the different batches do not have significant overlap in their state-action visitation frequencies. This is illustrated in Fig. 1.

## 3 Proposed algorithm

### 3.1 Learning multi-task policy from offline data with distillation

In Multi-task RL, [14–18] demonstrate the success of distilling multiple single-task policies into a multi-task policy. Inspired by these works, we propose a distillation procedure to obtain a multi-task policy in the Multi-task Batch RL setting. In Sec. 3.2, we argue such distillation procedure alone is insufficient due to the constraints the batch setting imposes on the policy search procedure.

The distillation procedure has two phases. In the first phase, we use BCQ to learn a different policy for each task, i.e. we learn $K$ different and independent policies. While we can use any Batch RL algorithm in the first phase, we use BCQ due to its simplicity. As described in Sec. 2.1, for each training batch, BCQ learns three functions: a state-action value function $Q$, a candidate action generator $G$ and a perturbation generator $\xi$. The output of the first phase thus consists of three sets of networks $\{Q_i\}_{i=1}^K$, $\{G_i\}_{i=1}^K$, and $\{\xi_i\}_{i=1}^K$, where $i$ indexes over the training tasks.

In the second phase, we distill each set into a network by incorporating a task inference module. The distilled function should recover different task-specific function depending on the inferred task identity. To distill the value functions $\{Q_i\}_{i=1}^{K}$ into a function $Q_D$, for each task $i$, we sample a context $\mathbf{c}_i$ and a pair $(s, a)$ from the batch $\mathcal{B}_i$. The task inference module $q_\phi$ takes $\mathbf{c}_i$ as input and infers a task identity $\mathbf{z}_i$. Given $\mathbf{z}_i$ as input, $Q_D$ should assign similar value to $(s, a)$ as the value function for the $i^{th}$ task $Q_i(s, a)$. The loss function with a $\beta$-weighted KL term [11] is:

$$\mathcal{L}_Q = \frac{1}{K} \sum_{i=1}^{K} \mathop{\mathbb{E}}_{(s,a),\mathbf{c}_i \sim \mathcal{B}_i} \left[ (Q_i(s,a) - Q_D(s,a,\mathbf{z}_i))^2 + \beta \text{KL}(q_\phi(\mathbf{c}_i) || \mathcal{N}(0,1)) \right], \quad \mathbf{z}_i \sim q_\phi(\mathbf{c}_i) \quad (3)$$

We also use Eq. 3 to train $q_\phi$ using the reparam trick [19]. Similarly, we distill the candidate action generators $\{G_i\}_{i=1}^{K}$ into $G_D$. $G_D$ takes as input state $s$, random noise $\nu$ and task identity $\mathbf{z}_i$. Depending on $\mathbf{z}_i$'s value, we train $G_D$ to regress towards the different candidate action generator:

$$\mathcal{L}_G = \frac{1}{K} \sum_{i=1}^{K} \mathop{\mathbb{E}}_{\substack{s,\mathbf{c}_i \sim \mathcal{B}_i \\ \nu \sim \mathcal{N}(0,1)}} \left[ ||G_i(s,\nu) - G_D(s,\nu,\bar{\mathbf{z}}_i)||^2 \right], \quad \mathbf{z}_i \sim q_\phi(\mathbf{c}_i). \quad (4)$$

The bar on top of $\bar{\mathbf{z}}_i$ in Eq. 4 indicates the stop gradient operation. We thus do not use the gradient of Eq. 4 to train the task inference module [11]. Lastly, we distill the perturbation generators $\{\xi_i\}_{i=1}^{K}$ into a single network $\xi_D$ (Eq. 5). $\xi_D$ takes as input a state $s$, a candidate action $a$, and an inferred task identity $\mathbf{z}_i$. We train $\xi_D$ to regress towards the output of $\xi_i$ given the same state $s$ and candidate action $a$ as input. We obtain the candidate action $a$ by passing $s$ through the candidate action generator $G_i$.

$$\mathcal{L}_\xi = \frac{1}{K} \sum_{i=1}^{K} \mathop{\mathbb{E}}_{\substack{s,\mathbf{c}_i \sim \mathcal{B}_i \\ \nu \sim \mathcal{N}(0,1)}} \left[ ||\xi_i(s,a) - \xi_D(s,a,\bar{\mathbf{z}}_i)||^2 \right], \quad \mathbf{z}_i \sim q_\phi(\mathbf{c}_i), \quad a = G_i(s,\nu) \quad (5)$$

Note that the gradient of $\mathcal{L}_\xi$ also updates $G_i$. The final distillation loss is given in Eq. 6. We parameterize $q_\phi, Q_D, G_D, \xi_D$ with feedforward NN as detailed in Appendix C.1.

$$\mathcal{L}_{distill} = \mathcal{L}_Q + \mathcal{L}_G + \mathcal{L}_\xi. \quad (6)$$

## 3.2 Robust task inference with triplet loss design

Given the high performance of distillation in Multi-task RL [14–18], it surprisingly performs poorly in Multi-task Batch RL, even on the training tasks. This is even more surprising because we can minimize the distillation losses (Fig. 2 top) and the single-task BCQ policies have high performance (Fig. 2 bottom). If the single-task policies perform well and we can distill them into a multi-task policy, why does the multi-task policy have poor performance? We argue the task inference module has learnt to model the posterior over task identity as conditionally dependent on only the state-action pairs in the context set , i.e. $P(Z|S, A)$, where $S, A$ are random variables denoting states and actions, rather than the correct dependency $P(Z|S, A, R)$ where $R$ denotes the rewards.

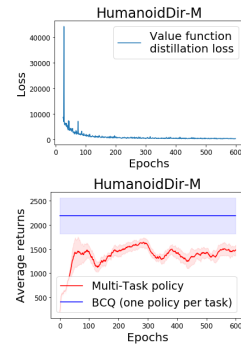

Figure 2: **Top:** Value function distillation loss (Eq. 3) during training. **Bottom:** The performance of the multi-task policy trained with Eq. 6 versus BCQ.

The behavior of the trained multi-task policy supports this argument. In this experiment, each task corresponds to a running direction. To maximize returns, the policy should run with maximal velocity in the target direction. We found that the multi-task policy often runs in the wrong target direction, indicating incorrect task inference. At the beginning of evaluation, the task identity is not provided. The policy takes random actions, after which it uses the collected transitions to infer the task identity. Having learnt the wrong conditional dependency, the task inference module assigns high probability mass in the posterior to region in the task embedding space whose training batches overlap with the collected transitions (Fig. 1).

The fundamental reason behind the wrong dependency is the non-overlapping nature of the training batches. Minimizing the distillation loss does not require the policy to learn the correct but more

**Algorithm 1** Distillation and triplet loss

**Input**: Batches $\{\mathcal{B}_i\}_{i=1}^K$; BCQ-trained $\{Q_i\}_{i=1}^K$, $\{G_i\}_{i=1}^K$, and $\{\xi_i\}_{i=1}^K$; randomly initialized $Q_D$, $G_D$ and $\xi_D$ jointly parameterized by $\theta$; task inference module $q_\phi$ with randomly initialized $\phi$

1: **repeat**
2:     Sample context set $\mathbf{c}_i$ from $\mathcal{B}_i, \forall i$
3:     Obtain relabelled transitions $\mathbf{c}_{j \to i}$ according to Eq. 7 for all pair of task $i, j$
4:     Calculate $\mathcal{L}_{triplet}$ using Eq. 9
5:     Calculate $\mathcal{L}_Q, \mathcal{L}_G, \mathcal{L}_\xi$ using Eq. 3, 4, 5
6:     Calculate $\mathcal{L}$ using Eq. 10
7:     Update $\theta, \phi$ to minimize $\mathcal{L}$
8: **until** Done

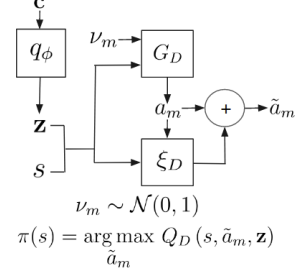

$$\nu_m \sim \mathcal{N}(0, 1)$$
$$\pi(s) = \arg\max_{\tilde{a}_m} Q_D(s, \tilde{a}_m, \mathbf{z})$$

Figure 3: Action selection. Given $s$, $G_D$ generates candidate actions $a_m$. $\xi_D$ generates small corrections for the actions $a_m$. The policy takes the corrected action $\tilde{a}_m$ with the highest value as estimated by $Q_D$.

complex dependency. The multi-task policy should imitate different single-task policy depending on which batch the context set was sampled from. If the batches do not overlap in state-action visitation frequencies, the multi-task policy can simply correlate the state-action pairs in the context with which single-task policy it should imitate. In short, if minimizing the training objective on the given datasets does not require the policy to model the dependency of the task identity on the rewards in the context set, there is no guarantee the policy will model this dependency. This is not surprising given literature on the non-identifiability of causality from observations [20, 21]. They also emphasize the benefit of using distribution change as training signal to learn the correct causal relationship [22].

Inspired by this literature, we introduce a distribution change into our dataset by approximating the reward function of each task $i$ with a learned function $\hat{R}_i$ (training illustrated in Appendix D). Given a context set $\mathbf{c}_j$ from task $j$, we relabel the reward of each transition in $\mathbf{c}_j$ using $\hat{R}_i$. Let $t$ index the transitions and $\mathbf{c}_{j \to i}$ denote the set of the relabelled transitions, we illustrate this process below [3] :

$$\mathbf{c}_j = \left\{ \left( s_{j,t}, a_{j,t}, r_{j,t}, s'_{j,t} \right) \right\}_t \xrightarrow{\text{Relabelling}} \mathbf{c}_{j \to i} = \left\{ \left( s_{j,t}, a_{j,t}, \hat{R}_i(s_{j,t}, a_{j,t}), s'_{j,t} \right) \right\}_t \quad (7)$$

Given the relabelled transitions, we leverage the triplet loss from the metric learning community [23] to enforce robust task inference, which is the most important design choice in MBML. Let $K$ be the number of training tasks, $\mathbf{c}_i$ be a context set for task $i$, $\mathbf{c}_j$ be a context set for task $j$ ($j \neq i$) , and $\mathbf{c}_{j \to i}$ be the relabelled set as described above, the triplet loss for task $i$ is:

$$\mathcal{L}_{triplet}^i = \frac{1}{K-1} \sum_{j=1, j \neq i}^K \left[ \underbrace{d\big(q_\phi\left(\mathbf{c}_{j \to i}\right), q_\phi\left(\mathbf{c}_i\right)\big)}_{\substack{\text{Ensure } \mathbf{c}_{j \to i} \text{ and } \mathbf{c}_i \text{ infer} \\ \textit{similar} \text{ task identities}}} - \underbrace{d\big(q_\phi\left(\mathbf{c}_{j \to i}\right), q_\phi\left(\mathbf{c}_j\right)\big)}_{\substack{\text{Ensure } \mathbf{c}_{j \to i} \text{ and } \mathbf{c}_j \text{ infer} \\ \textit{different} \text{ task identities}}} + m \right]_+, \quad (8)$$

where $m$ is the triplet margin, $[\cdot]_+$ is the ReLU function and $d$ is a divergence measure. $q_\phi$ outputs the posterior over task identity, we thus choose $d$ to be the KL divergence.

Minimizing Eq. 8 accomplishes two goals. It encourages the task inference module $q_\phi$ to infer similar task identities when given either $\mathbf{c}_i$ or $\mathbf{c}_{j \to i}$ as input. It also encourages $q_\phi$ to infer different task identities for $\mathbf{c}_j$ and $\mathbf{c}_{j \to i}$. We emphasize that the task inference module can not learn to correlate *only* the state-action pairs with the task identity since $\mathbf{c}_j$ and $\mathbf{c}_{j \to i}$ contain the same state-action pairs, but they correspond to different task identities. To minimize Eq. 8, the module must model the correct conditional dependency $P(Z|S, A, R)$ when inferring the task identity.

Eq. 8 calculates the triplet loss when we use the learned reward function of task $i$ to relabel transitions from the remaining tasks. Following similar procedures for the remaining tasks lead to the loss:

$$\mathcal{L}_{triplet} = \frac{1}{K} \sum_{i=1}^K \mathcal{L}_{triplet}^i. \quad (9)$$

The final loss to train the randomly initialized task inference module $q_\phi$, the distilled value functions $Q_D$, the distilled candidate action generator $G_D$, and the distilled perturbation generator $\xi_D$ is:

$$\mathcal{L} = \mathcal{L}_{triplet} + \mathcal{L}_Q + \mathcal{L}_G + \mathcal{L}_\xi. \tag{10}$$

Alg. 1 illustrates the pseudo-code for the second phase of the distillation procedure. Detailed pseudo-code of the two-phases distillation procedures can be found in Appendix E. Fig. 3 briefly describes action selection from the multi-task policy. Appendix F provides detailed explanations. In theory, we can also use the relabelled transitions in Eq. 7 to train the single-task BCQ policy in the first phase, which we do not since we focus on task inference in this work.

## 4   Discussions

The issue of learning the wrong dependency does not surface when multi-task policies are tested in Atari tasks because their state space do not overlap [18, 24, 25]. Each Atari task has distinctive image-based state. The policy can perform well even when it only learns to correlate the state to the task identity. When Mujoco tasks are used to test online multi-task algorithms [13, 26], the wrong dependency becomes self-correcting. If the policy infers the wrong task identity, it will collect training data which increases the overlap between the datasets of the different training tasks, correcting the issue overtime. However, in the batch setting, the policy can not collect more transitions to self-correct inaccurate task inference. Our insight also leads to exciting possibility to incorporate mechanism to quickly infer the correct causal relationship and improve sample efficiency in Multi-task RL, similar to how causal inference method has motivated new innovations in imitation learning [27].

Our first limitation is the reliance on the generalizability of simple feedforward NN. Future research can explore more sophisticated architecture, such as Graph NN with reasoning inductive bias [28–31] or structural causal model [32, 33], to ensure accurate task inference. We also assume the learnt reward function of one task can generalize to state-action pairs from the other tasks, even when their state-action visitation frequencies do not overlap significantly. To increase the prediction accuracy, we use a reward ensemble to estimate epistemic uncertainty (Appendix D). We note that the learnt reward functions do not need to generalize to every state-action pairs, but only enough pairs so that the task inference module is forced to consider the rewards when trained to minimize Eq. 8. Crucially, we do not need to solve the task inference challenge while learning the reward functions and using them for relabelling, allowing us to side-step the challenge of task inference.

The second limitation is in scope. We only demonstrate our results on tasks using proprioceptive states. Even though they represent high-dimensional variables in a highly nonlinear ODE, the model does not need to tackle visual complexity. The tasks we consider also have relatively dense reward functions and not binary reward functions. These tasks, such as navigation and running, are also quite simple in the spectrum of possible tasks we want an embodied agents to perform. These limitations represent exciting directions for future work.

Another interesting future direction is to apply supervised learning self-distillation techniques [34, 35], proven to improve generalization, to further improve the distillation procedure. To address the multi-task learning problem for long-horizon tasks, it would also be beneficial to consider skill discovery and composition from the batch data [36, 37]. However, in this setting, we still need effective methods to infer the correct task identity to perform well in unseen tasks. Our explanation in Sec. 3 only applies when the tasks differ in reward function. Extending our approach to task distributions with varying transition functions is trivial. Sec. 5 provide experimental results for both cases.

## 5   Experiment Results

We demonstrate the performance of our proposed algorithm (Sec. 5.1) and ablate the different design choices (Sec. 5.2). Sec. 5.3 shows that the multi-task policy can serve as a good initialization, significantly speeding up training on unseen tasks. Appendix C provides all hyper-parameters.

### 5.1   Performance evaluation on unseen tasks

We evaluate in five challenging task distributions from MuJoCo [38] and a modified task distribution UmazeGoal-M from D4RL [39]. In AntDir and HumanoidDir-M, a target direction defines a task.

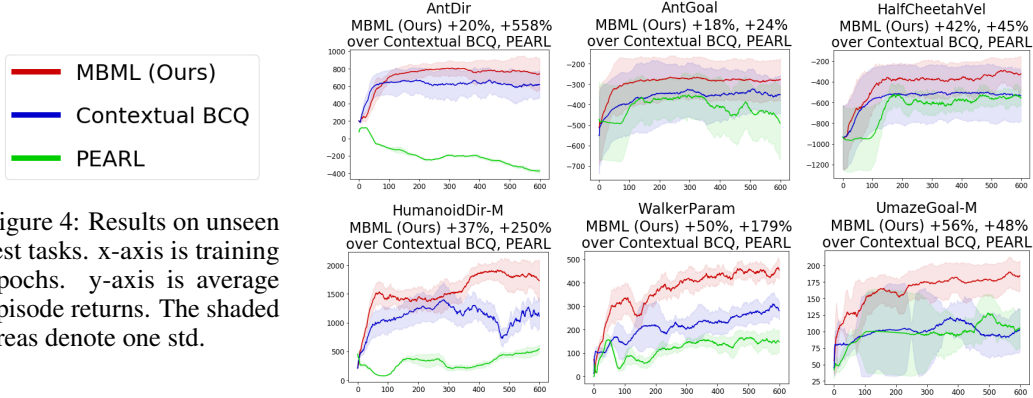

Figure 4: Results on unseen test tasks. x-axis is training epochs. y-axis is average episode returns. The shaded areas denote one std.

The agent maximizes returns by running with maximal speed in the target direction. In AntGoal and UmazeGoal-M, a task is defined by a goal location, to which the agent should navigate. In HalfCheetahVel, a task is defined as a constant velocity the agent should achieve. We also consider the WalkerParam environment where random physical parameters parameterize the agent, inducing different transition functions in each task. The state for each task distribution is the OpenAI gym state. We do not include the task-specific information, such as the goal location or the target velocity in the state. The target directions and goals are sampled from a $120°$ circular arc. Details of these task distributions can be found in Appendix H.1.

We argue that the version of HumanoidDir used in prior works does not represent a meaningful task distribution, where a single task policy can already achieve the optimal performance on unseen tasks. We thus modify the task distribution so that a policy has to infer the task identity to perform well, and denote it as HumanoidDir-M. More details of this task distribution can be found in Appendix G.

There are two natural baselines. The first is by modifying PEARL [11] to train from the batch, instead of allowing PEARL to collect more transitions. We thus do not execute line $1 - 10$ in Algorithm 1 in the PEARL paper. On line 13, we sample the context and the RL batch uniformly from the batch. The second baseline is Contextual BCQ. We modify the networks in BCQ to accept the inferred task identity as input. We train the task inference module using the gradient of the value function loss. MBML and the baselines have the same network architecture. We are very much inspired by PEARL and BCQ. However, we do not expect PEARL to perform well in our setting because it does not explicitly handle the difficulties of learning from a batch without interactions. We also expect that our proposed algorithm will outperform Contextual BCQ thanks to more robust task inference.

We measure performance by the average returns over unseen tasks, sampled from the same task distribution. We do not count the first two episodes' returns [11]. We obtain the batch for each training task by training Soft Actor Critic (SAC) [40] with a fixed number of environment interactions. Appendix H provide more details on the environment setups and training procedures of the baselines.

From Fig. 4, MBML outperforms the baselines by a healthy margin in all task distributions. Even though PEARL does not explicitly handle the challenge of training from an offline batch, it is remarkably stable, only diverging in AntDir. Contextual BCQ is stable, but converges to a lower performance than MBML in all task distributions. An astude reader will notice the issue of overfitting, for example Contextual BCQ in HumanoidDir-M. Since our paper is not about determining early stopping conditions and to ensure fair comparisons among the different algorithms, we compute the performance comparisons using the best results achieved by each algorithm during training.

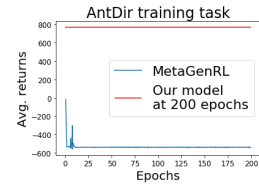

Figure 5: MetaGenRL quickly diverges and does not recover.

We also compare with MetaGenRL [41]. Since it relies on DDPG [42] to estimate value functions, which diverges in Batch RL [9], we do not expect it to perform well in our setting. Fig. 5 confirms this, where its performance quickly plummets and does not recover with more training. Combining MetaGenRL and MBML is interesting since MetaGenRL generalizes to out-of-distribution tasks.

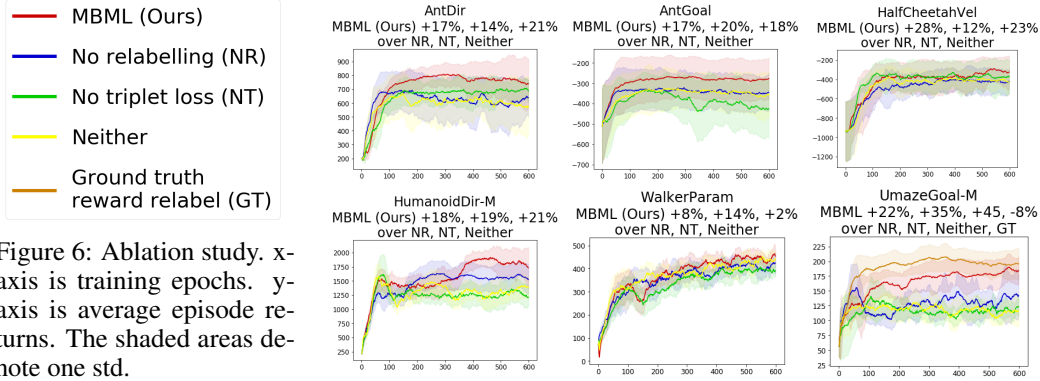

Figure 6: Ablation study. x-axis is training epochs. y-axis is average episode returns. The shaded areas denote one std.

## 5.2 Ablations

We emphasize that our contributions lie in the triplet loss design coupled with transitions relabelling. Below, we provide ablation studies to demonstrate that both are crucial to obtain superior performance.

**No relabelling.** To obtain hard negative examples, we search over a mini-batch to find the hardest positive-anchor and negative-anchor pairs, a successful and strong baseline from metric learning [23]. This requires sampling $N$ context sets $\{c_i^n\}_{n=1}^N$ for each task $i$, where $n$ indexes the context sets sampled for each task. Let $K$ be the number of training tasks, the triplet loss is:

$$\frac{1}{K} \sum_{i=1}^{K} \left[ \max_{n,n'=1,...,N} d\left(q_\phi(\mathbf{c}_i^n), q_\phi(\mathbf{c}_i^{n'})\right) - \min_{\substack{n,n'=1,...,N \\ j=1,...,K, j\neq i}} d\left(q_\phi(\mathbf{c}_i^n), q_\phi(\mathbf{c}_j^{n'})\right) + m \right]_+ . \quad (11)$$

The $max$ term finds the positive-anchor pair for task $i$ by considering every pair of context sets from task $i$ and selecting the pair with the largest divergence in the posterior over task identities. The $min$ term finds the negative-anchor pair for task $i$ by considering every possible pair between the context sets sampled for task $i$ and the context sets sampled for the other tasks. It then selects the pair with the lowest divergence in the posterior over task identities as the negative-anchor pair.

**No triplet loss.** We train the task inference module using only gradient of the value function distillation loss (Eq. 3). To use the relabelled transitions, the module also takes as input the relabelled transitions during training. More concretely, given the context set $\mathbf{c}_i$ from task $i$, we sample an equal number of relabelled transitions from the other tasks $\tilde{\mathbf{c}}_i \sim \cup_j \mathbf{c}_{j\rightarrow i}$. During training, the input to the task inference module is the union of the context set $\mathbf{c}_i$ and the sampled relabelled transitions $\tilde{\mathbf{c}}_i$. In the full model, we also perform similar modification to the input of the module during training.

**No transition relabelling and no triplet loss.** This method is a simple combination of a task inference module and the distillation process. We refer to this algorithm as **Neither** in the graphs.

Fig. 6 compares our full model and the ablated versions. Our full model obtains higher returns than most of the ablated versions. For WalkerParam, our full model does not exhibit improvement over **Neither**. However, from Fig. 4, our full model significantly outperforms the baselines. We thus conclude that, in WalkerParam, the improvement over the baselines comes from distillation.

Comparing to the **No triplet loss** ablation, transition relabelling leads to more efficient computation of the triplet loss. Without the relabelled transitions, computing Eq. 11 requires $O(K^2N^2)$. Our loss in Eq. 9 only requires $O(K^2)$. We also need to relabel the transitions only once before training the multi-task policy. It is also trivial to parallelize across tasks.

We also study reward estimation accuracy. Fig. 7 shows that our reward model achieves low error on state-action pairs from another task, both with and without an ensemble. We also compare MBML against an ablated version that uses the ground truth reward function for relabelling on UmazeGoal-M. The model trained using the ground truth reward function

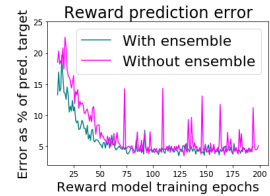

Figure 7: Error on unseen task.

only performs slightly better than the model trained using the learned reward function. We include in Appendix I experiments on margin sensitivity analysis and the benefit of the reward ensemble.

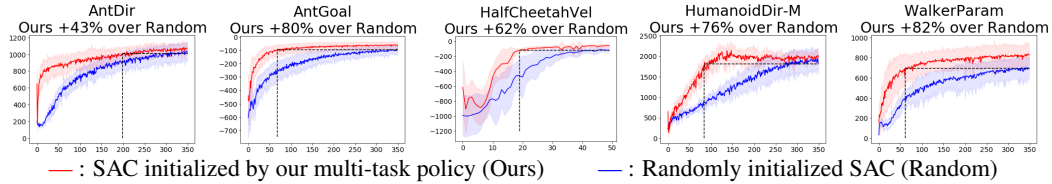

Figure 8: Initialization results. x-axis is number of interactions in thousands. y-axis is the average episode returns over unseen tasks. The shaded areas denote one std.

## 5.3 Using the multi-task policy to enable faster convergence when training on unseen tasks

While the multi-task policy generalize to unseen tasks, its performance is not optimal. If we allow further training, initializing networks with our multi-task policy significantly speeds up convergence to the optimal performance.

The initialization process is as followed. Given a new task, we use the multi-task policy to collect 10K transitions. We then train a new policy to imitate the actions taken by maximizing their log likelihood. As commonly done, the new policy outputs the mean and variance of a diagonal Gaussian distribution. The new policy does not take a task identity as input. The task inference module infers a task identity $\mathbf{z}$ from the 10K transitions. Fixing $\mathbf{z}$ as input, the distilled value function $Q_D$ initializes the new value function. Given the new policy and the initialized value function, we train them with SAC by collecting more data. To stabilize training, we perform target policy smoothing [43] and double-Q learning [44] by training two identically initialized value functions with different mini-batches (pseudo-codes and more motivations in Appendix J.1).

Fig. 8 compares the performance of the policies initialized with our multi-task policy to randomly initialized policies. Initializing the policies with the MBML policy significantly increases convergence speed in all five task distributions, demonstrating our method's robustness. Even in the complex HumanoidDir-M task distribution, our method significantly speeds up the convergence, requiring only 85K environment interactions, while the randomly initialized policies require 350K, representing a 76% improvement in sample efficiency. Similar conclusions hold when comparing against randomly initialized SAC where the two value functions are trained using different mini-batches (Appendix J.2). We also note that our initialization method does not require extensive hyper-parameter tuning.

## 6 Related Works

**Batch RL** Recent advances in Batch RL [7–10, 45] focus on the single-task setting, which does not require training a task inference module. Thus they are not directly applicable to the Multi-task Batch RL. [6, 46] also consider the multi-task setting but assume access to the ground truth task identity and reward function of the test tasks. Our problem setting also differs, where the different training batches do not have significant overlap in state-action visitation frequencies, leading to the challenge of learning a robust task inference module.

**Task inference in multi-task setting** The challenge of task inference in a multi-task setting has been tackled under various umbrellas. Meta RL [11, 13, 26, 47–50] trains a task inference module to infer the task identity from a context set. We also follow this paradigm. However, our setting presents additional challenge to train a robust task inference module, which motivates our novel triplet loss design. As the choice of loss function is crucial to train an successful task inference module in our settings, we will explore the other loss functions, e.g. loss functions discussed in [51], in future work. Other multi-task RL works [52–55] focus on training a good multi-task policy, rather than the task inference module, which is an orthogonal research direction to ours.

**Meta RL** Meta RL [48, 56–60] optimizes for quick adaptation. However, they require interactions with the environment during training. Even though we do not explicitly optimize for quick adaptation, we demonstrate that initializing a model-free RL algorithm with our policy significantly speeds up convergence on unseen tasks. [26] uses the data from the training tasks to speed up convergence when learning on new tasks by propensity estimation techniques. This approach is orthogonal to ours and can potentially be combined to yield even greater performance improvement.

## Acknowledgement

We acknowledge Professor Keith Ross (NYU) for initial discussions and inspiration for this work. We thank Fangchen Liu (UC Berkeley) for pointing out a figure issue right before the paper submission deadline. We thank Chaochao Lu (University of Cambridge) for introducing us to causality. Computing needs were supported by the Nautilus Pacific Research Platform.

## Broader Impact

### Positive impact

Our work provides a solution to learn a policy that generalizes to a set of similar tasks from only observational data. The techniques we propose have great potential to benefit various areas of the whole society. For example in the field of healthcare, we hope the proposed triplet loss design with hard negative mining can enable us to robustly train an automatic medical prescription system from a large batch of medical histories of different diseases and further generalize to new diseases [61], e.g., COVID-19. Moreover, in the field of robotics, our methods can enable the learning of a single policy that solves a set of similar unseen tasks from only historical robot experiences, which tackles the sample efficiency issues given that sampling is expensive in the field of real-world robotics [46]. Even though in some fields that require safe action selections, e.g, autonomous driving [62] and medical prescription, our learned policy cannot be immediately applied, it can still serve as a good prior to accelerate further training.

### Negative impact

Evidently, the algorithm we proposed is a data-driven methods. Therefore, it is very likely that it will be biased by the training data. Therefore, if the testing tasks are very different from the training tasks, the learned policy may even result in worse behaviors than random policy, leading to safety issues. This will motivate research into safe action selection and distributional shift identification when learning policies for sequential process from only observational data.

## Funding Disclosure

Nothing to disclose.

## Footnotes

[3]The idea of modeling the reward function and using that model to relabel the rewards from the batches originally came from Professor Keith Ross.

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
