[Supplementary Material]

# Appendices

## A Symbol definition

Table 1: Symbol definition. Some of the symbol are overloaded. We make sure each term is clearly defined given the context.

| Symbol | Definition | Dimension |
|---|---|---|
| $\mathcal{S}$ | state space | $\mathbb{R}^{N_s}$ |
| $\mathcal{A}$ | action space | $\mathbb{R}^{N_a}$ |
| $T$ | transition function | $\mathbb{R}^{N_s+N_a} \to \mathbb{R}^{N_s}$ |
| $T_0$ | initial state distribution | $\mathbb{R}^{N_s} \to [0,1]$ |
| $R$ | reward function | $\mathbb{R}^{N_s+N_a+N_s} \to \mathbb{R}$ |
| $H$ | horizon | $\mathbb{N}_+$ |
| $M$ | MDP $M = (\mathcal{S}, \mathcal{A}, T, T_0, R, H)$, which defines a task | - |
| $p(M)$ | task distribution | - |
| $N_\theta$ | dimension of the policy parameter | $\mathbb{N}_+$ |
| $K$ | number of task | $\mathbb{N}_+$ |
| $\tau_M$ | trajectory generated by interacting with $M$ | - |
| $s_t$ | state at time step $t$ | $\mathbb{R}^{N_s}$ |
| $a_t$ | action selected at time step $t$ | $\mathbb{R}^{N_a}$ |
| $\tilde{a}$ | corrected action | $\mathbb{R}^{N_a}$ |
| $s_t'$ | state at time step $t$ | $\mathbb{R}^{N_s}$ |
| $\pi_\theta$ | policy function, parameterized by $\theta$ | $\mathbb{R}^{N_s} \to \mathbb{R}^{N_a}$ |
| $N$ | number of transition tuples from one task batch | $\mathbb{N}_+$ |
| $\mathcal{B}_i$ | batch of transition tuples for task $i$ | - |
| $\hat{R}_i$ | learned reward function for task $i$ | $\mathbb{R}^{N_s+N_a+N_s} \to \mathbb{R}$ |
| $N_c$ | number of transition tuples in a context set | $\mathbb{N}_+$ |
| $\mathbf{c}_i$ | context set for task $i$ | $\mathbb{R}^{(N_s+N_a+N_s)*N_c}$ |
| $\mathbf{c}_{j \to i}$ | relabeled context set by uing $\hat{R}_i$ | $\mathbb{R}^{(N_s+N_a+N_s)*N_c}$ |
| $\tilde{\mathbf{c}}_i$ | Union of relabeled context set by uing $\hat{R}_i$ | $\mathbb{R}^{(N_s+N_a+N_s)*N_c*(K-1)}$ |
| $\mathbf{z}_i$ | task identity for task $i$ | $\mathbb{R}^{N_z}$ |
| $q_\phi$ | task inference module, parameterized by $\phi$ | $\mathbb{R}^{(N_s+N_a+N_s)*N_c} \to (N_z, N_z)$ |
| $S, A, R, Z$ | random variables: states, actions, rewards, task identity | - |
| $J_{M_i}(\pi_\theta)$ | expected sum of rewards in $M_i$ induced by policy $\pi_\theta$ | $\mathbb{R}^{N_\theta} \to \mathbb{R}$ |
| $J(\pi_\theta)$ | expected sum of rewards in $p(M)$ induced by policy $\pi_\theta$ | $\mathbb{R}^{N_\theta} \to \mathbb{R}$ |
| $Q$ | Q value function | $\mathbb{R}^{N_s+N_a} \to \mathbb{R}$ |
| $Q_i$ | Q value function for task $i$ | $\mathbb{R}^{N_s+N_a} \to \mathbb{R}$ |
| $Q_D$ | distilled Q value function | $\mathbb{R}^{N_s+N_a+N_z} \to \mathbb{R}$ |
| $G$ | candidate action generator | $\mathbb{R}^{N_s+1} \to \mathbb{R}^{N_a}$ |
| $G_i$ | candidate action generator for task $i$ | $\mathbb{R}^{N_s+1} \to \mathbb{R}^{N_a}$ |
| $G_D$ | distilled candidate action generator | $\mathbb{R}^{N_s+1+N_z} \to \mathbb{R}^{N_a}$ |
| $\xi$ | perturbation generator | $\mathbb{R}^{N_s+N_a} \to \mathbb{R}^{N_a}$ |
| $\xi_i$ | perturbation generator for task $i$ | $\mathbb{R}^{N_s+N_a} \to \mathbb{R}^{N_a}$ |
| $\xi_D$ | distilled perturbation generator | $\mathbb{R}^{N_s+N_a+N_z} \to \mathbb{R}^{N_a}$ |
| $\mathcal{N}(0,1)$ | standard Gaussian distribution | - |
| $\nu$ | noise sampled from standard Gaussian distribution | $\mathbb{R}$ |
| $m$ | triplet margin | $\mathbb{R}$ |
| $d(\cdot)$ | divergence measure | - |
| KL | KL divergence | - |
| $\bar{\cdot}$ | stop gradient operation | - |
| $\mathcal{L}_Q$ | loss function to distill $Q_D$ | - |
| $\mathcal{L}_G$ | loss function to distill $G_D$ | - |
| $\mathcal{L}_\xi$ | loss function to distill $\xi_D$ | - |
| $\mathcal{L}_{distill}$ | total distillation loss | - |
| $\mathcal{L}^i_{triplet}$ | triplet loss for task $i$ | - |
| $\mathcal{L}_{triplet}$ | mean triplet loss across all tasks | - |
| $\mathcal{L}$ | final loss: $\mathcal{L} = \mathcal{L}_{triplet} + \mathcal{L}_{distill}$ | - |

# B    Action selection of BCQ policy

$$\pi\left(s\right) = \underset{a_m + \xi(s, a_m)}{\arg\max}\; Q\left(s, a_m + \xi\left(s, a_m\right)\right), \qquad \left\{a_m = G\left(s, \nu_m\right)\right\}_m, \qquad \nu_m \sim \mathcal{N}(0, 1).$$

Figure 9: Action selection procedure of BCQ.

In this section, we provide the detailed action selection procedures for BCQ. To pick action given a state $s$, we first sample a set of small noises $\{\nu_m\}_m$ from the standard Gaussian distribution. For each $\nu_m$, the candidate action generator $G$ will generate a candidate action $a_m$ for state $s$. For each of the candidate actions $a_m$, the perturbation model $\xi$ will generate a small correction term $\xi(s, a_m)$ by taking as input the state-candidate action pair. Therefore, a set of corrected candidate actions $\{a_m + \xi(s, a_m)\}_m$ will be generated for the state $s$. The corrected candidate action with the highest estimated $Q$ value will be selected as $\pi\left(s\right)$.

# C Hyper-parameters

## C.1 Hyper-parameters of our proposed models

Table 2: Hyper-parameters of our proposed model

| Hyper-parameters | Value |
|---|---|
| Number of evaluation episodes | 5 |
| Task identity dimension | 20 |
| Number of candidate actions | 10 |
| Learning rate | 0.0003 |
| Training batch size | 128 |
| Context set size | 64 |
| KL regularization weighting term $\beta$ | 0.1 |
| Triplet margin $m$ | 2.0 |
| Reward prediction ensemble $\sigma_{\text{threshold}}$ | AntDir, AntGoal: 0.1<br>WalkerParam: 0.1<br>HumanoidDir-M: 0.2<br>HalfCheetahVel: 0.05<br>UmazeGoal-M: 0.02 |
| Next state prediction ensemble $\sigma_{\text{threshold}}$ | 0.1 |
| $Q_D$ architecture | MLP with 9 hidden layers, 1024 nodes each, ReLU activation |
| $G_D$ architecture | MLP with 7 hidden layers, 1024 nodes each, ReLU activation |
| $\xi_D$ architecture | MLP with 8 hidden layers, 1024 nodes each, ReLU activation |

Table 3: Hyper-parameters of reward and next state prediction ensemble

| Hyper-parameters | Value |
|---|---|
| Learning rate | 0.0003 |
| Training batch size | 128 |
| Reward prediction ensemble size | 20 |
| Reward prediction network architecture | MLP with 1 hidden layers, 128 nodes, ReLU activation |
| Next state prediction ensemble size | 20 |
| Next state prediction network architecture | MLP with 6 hidden layers, 256 nodes each, ReLU activation |

Table 2 provides the hyper-parameters for our proposed model and all of its ablated versions (Sec. 3, Sec. 5.2). The hyper-parameters for the reward ensembles and next state prediction ensembles are provided in Table 3. Our model uses the task inference module from PEARL with the same architecture, described in Table 4. Since the scale of the reward in different task distributions are different, we need to use different values for the reward prediction ensemble threshold $\sigma_{\text{threshold}}$.

We did not conduct extensive search to determine the hyper-parameters. Instead, we reuse some default hyper-parameter settings from the other multi-task learning literature on the MuJoCo benchmarks [11, 26]. As for the architecture of the distillation networks, we select reasonably deep networks.

When using BCQ to train the single-task policies in the first phase of the distillation procedure, we use the default hyper-parameters in the official implementation of BCQ, except for the learning rate, which decreases from 0.001 to 0.0003. We find lowering the learning rate leads to more stable learning for BCQ.

## C.2 Hyper-parameters of Contextual BCQ

For Contextual BCQ, the value function, decoder, and perturbation model have the same architecture as $Q_D, G_D, \xi_D$ in our model. The encoder also has the same architecture as the decoder. The task inference module has the same architecture as the task inference module in PEARL, described in Table 4.

The context set size used during training Contextual BCQ is 128, twice the size of the context set in our model. This is because during training of our model, we use the combination of context transitions and the same number of relabelled transitions from the other tasks to infer the posterior over task identity, as detailed in Sec. 5.2 and pseudo-codes provided in Alg. 4. Therefore, the effective number of transitions that are used as input into the task inference module during training are the same for our model and Contextual BCQ.

Unless stated otherwise, for the remaining hyper-parameters, such as the maximum value of the perturbation, we use the default value in BCQ.

### C.3 Hyper-parameters of PEARL

Table 4: Hyper-parameters of PEARL

| Hyper-parameters | Value |
|---|---|
| Task inference module architecture | MLP with 3 hidden layers, 200 nodes each, ReLU activation |

We use the default hyper-parameters as provided in the official implementation of PEARL. For completeness when discussing the hyper-parameters of our model, we provide the architecture of the task inference module in Table 4.

### C.4 Hyper-parameters of ablation studies of the full model

Table 5: Hyper-parameters of **No transition relabelling**

| Hyper-parameters | Value |
|---|---|
| Number of sampled context sets $N$ | 10 |
| Context set size | 128 |

Table 6: Hyper-parameters of **No triplet loss**

| Hyper-parameters | Value |
|---|---|
| Context set size | 64 |

Table 7: Hyper-parameters of **Neither**

| Hyper-parameters | Value |
|---|---|
| Context set size | 128 |

Table 5, Table 6 and Table 7 provide the hyper-parameters for the ablated versions of our full model **No transition relabelling**, **No triplet loss**, and **Neither**, respectively. Without the transition relabelling techniques, **No transition relabelling** and **Neither** set the size of training context size to 128 as Contextual BCQ to use the same effective number of transitions to infer the posterior over the task identity as our full model. Note that the remaining hyper-parameters of these methods are set to be the same as our full model, described in Table 2.

### C.5 Hyper-parameters when we initialize SAC with our multi-task policy

Table 8: Hyper-parameters of SAC when initialized by our multi-task policy

| Hyper-parameters | Value |
|---|---|
| Q function architecture | MLP with 9 hidden layers, 1024 nodes each, ReLU activation |
| Q function target smoothing rate | 0.005 |
| policy target smoothing rate | 0.1 |

The architecture of the Q function network is the same as the distilled Q function $Q_D$ in Table 2. The Q function target smoothing rate is the same as the standard SAC implementation [40]. The policy target smoothing rate is searched over $\{0.005, 0.01, 0.1, 0.5\}$. For the SAC trained from random initialization baseline (Appendix J.2), we also change the sizes of the value function to the same value in Table 8. For the remaining hyper-parameters, we use the default hyper-parameter settings of SAC.

# D    Reward prediction ensemble

---

**Algorithm 2** Training procedure of reward function approximator

---

**Input**: data batch $\mathcal{B}_i$; $\hat{R}_{i,l}$ with randomly initialized parameters.

  1: **for** a fixed number of iterations **do**
  2:      Sample a transition $(s, a, r, s')$ from $\mathcal{B}_i$
  3:      Obtain the predicted reward $\hat{r} = \hat{R}_{i,l}(s, a)$
  4:      Update parameters of $\hat{R}_{i,l}$ to minimize $(\hat{r} - r)^2$ through gradient descent.
  5: **end for**

**Output**: trained reward function approximator $\hat{R}_{i,l}$

---

---

**Algorithm 3** Relabel transition from task $j$ to task $i$

---

**Input**: an ensemble of learned reward functions $\{\hat{R}_{i,l}\}_l$; context set $\mathbf{c}_j = \left\{ \left( s_{j,t}, a_{j,t}, r_{j,t}, s'_{j,t} \right) \right\}_t$
from task $j$, a threshold $\sigma_{\text{threshold}}$.

  1: $\mathbf{c}_{j \to i} \leftarrow \{\}$
  2: **for** $t = 1, \ldots, |\mathbf{c}_j|$ **do**
  3:      **if** $\text{std}(\{\hat{R}_{i,l}(s_{j,t}, a_{j,t})\}_l) < \sigma_{\text{threshold}}$ **then**
  4:          $\hat{R}_i(s_{j,t}, a_{j,t}) \leftarrow \text{mean}(\{\hat{R}_{i,l}(s_{j,t}, a_{j,t})\}_l)$
  5:          Add $(s_{j,t}, a_{j,t}, \hat{R}_i(s_{j,t}, a_{j,t}), s'_{j,t})$ to $\mathbf{c}_{j \to i}$
  6:      **end if**
  7: **end for**

**Output**: relabelled transitions $\mathbf{c}_{j \to i}$

---

In subsection 3.2, we propose to train a reward function approximator $\hat{R}_i$ for each training task $i$ to relabel the transitions from the other tasks. To increase the accuracy of the estimated reward, for each task $i$, we use an ensemble of learnt reward functions $\{\hat{R}_{i,l}\}_l$, where $i$ indexes the task and $l$ indexes the function in the ensemble. The training procedures for each reward function approximator in the ensemble are provided in Alg. 2.

The pseudo-code for generating relabelled context set $\mathbf{c}_{j \to i}$ from context set $\mathbf{c}_j$ of task $j$ is given in Alg. 3. We use the output of the ensemble as an estimate of the epistemic uncertainty in the reward prediction [63]. Concretely, for each transition in $\mathbf{c}_j$, we only include it in the relabelled set $\mathbf{c}_{j \to i}$ if the standard deviations of the ensemble output is below a certain threshold (line 3). We also use the mean of the outputs as the estimated reward (line 4).

We conduct ablation study of the reward prediction ensemble in Appendix I.2, where we show that the use of reward prediction ensemble improves the performance when initializing SAC with our multi-task policy.

# E  Detailed pseudo-codes of the two-phases distillation procedures

In this section, we provide the detailed pseudo-code in Alg. 4 for the two-phases distillation procedures introduced in Sec. 3. The basic idea is that we first obtain single-task policy for each training task using BCQ. In the second phase, we distill the single-task policies into a multi-task policy by incorporating a task inference module. Note that the task inference module is trained by minimizing the Q value function distillation loss (Eq. 3) and the triplet loss (Eq. 9).

Line 1 describes the first phase of the two-phases distillation procedure. We use BCQ to learn a state-action value function $Q_i$, a candidate action generator $G_i$ and a perturbation generator $\xi_i$ for each training batch $\mathcal{B}_i$.

We next enter the second phase. We first sample context set $\mathbf{c}_i$ of size $N_{\text{context}}$ from $\mathcal{B}_i$, $i = 1, \ldots, K$ in line 3. Line 5-10 provide the procedures to calculate the triplet loss. For each task $i$, we relabel the reward of each transition in all the remaining context set $\mathbf{c}_j$ using $\hat{R}_i$ and obtain $\mathbf{c}_{j \to i}, \forall j \neq i$ in line 5. From the union of the relabelled context set $\cup_j \mathbf{c}_{j \to i}$, we sample a subset $\tilde{\mathbf{c}}_i$ of size $N_{\text{context}}$ in line 6. Denote transitions in $\tilde{\mathbf{c}}_i$ originated from $\mathbf{c}_j$ as $\mathbf{x}_{j \to i}$. Further denote transitions in $\mathbf{x}_{j \to i}$ before relabelling as $\mathbf{x}_j$, we thus have $\mathbf{x}_j \in \mathbf{c}_j$. These sets of transitions have the following relationships:

$$\cup_j \mathbf{c}_{j \to i} \xrightarrow{\text{Sample}} \tilde{\mathbf{c}}_i, \quad \tilde{\mathbf{c}}_i = \cup_j \mathbf{x}_{j \to i}$$

$$\mathbf{x}_{j \to i} \in \mathbf{c}_{j \to i}, \quad \mathbf{x}_j \xrightarrow{\text{Relabel}} \mathbf{x}_{j \to i} \tag{12}$$

To calculate the triplet loss for task $i$, in line 9 we sample a subset $\mathbf{c}_{i,j}$ with the same number of transitions as $\mathbf{x}_j$ from $\mathbf{c}_i$, i.e. $|\mathbf{c}_{i,j}| = |\mathbf{x}_j|$ for each $j \neq i$. Therefore, the triplet loss for task $i$ can be given by Eq. 13.

Line 11-13 provide the procedures to infer the task identity for each task $i$. We use the union of the context set $\mathbf{c}_i$ and the relabeled context set $\tilde{\mathbf{c}}_i$ sampled from $\cup_j \mathbf{c}_{j \to i}$ to infer the posterior $q_\phi(\mathbf{z} | \{\mathbf{c}_i, \tilde{\mathbf{c}}_i\})$ over task identity. We next sample the task identity $\mathbf{z}_i$ from $q_\phi(\mathbf{z} | \{\mathbf{c}_i, \tilde{\mathbf{c}}_i\})$.

To calculate the distillation loss of each distilled function, in line 14 we sample the training batch of $N$ transitions from $\mathcal{B}_i$. With $\mathbf{z}_i$ and the training transition batch, we can calculate the value function distillation loss $\mathcal{L}_Q^i$ of task $i$ using Eq. 14. To calculate the distillation loss of the candidate action generator $G_D$ and perturbation generator $\xi_D$ of task $i$, we first sample $N$ noises $\nu_t$ from the standard Gaussian distribution $\mathcal{N}(0, 1)$ in line 16. In line 17, we then obtain the candidate actions $\hat{a}_t = G_i(s_t, \nu_t)$ for each state $s_t$ in the training batch. The calculations to derive $\mathcal{L}_G^i$ and $\mathcal{L}_\xi^i$ for task $i$ follow Eq. 15 and Eq. 16, respectively.

After repeating the procedures for all the training tasks, in line 21-24 we average the losses across tasks and obtain $\mathcal{L}_{triplet}$, $\mathcal{L}_Q$, $\mathcal{L}_G$, and $\mathcal{L}_\xi$. At the end of each iteration, we update $\theta$ and $\phi$ by minimizing $\mathcal{L} = \mathcal{L}_{triplet} + \mathcal{L}_Q + \mathcal{L}_G + \mathcal{L}_\xi$ in line 25.

**Algorithm 4** Two-phases distillation procedure with novel triplet loss design

---

**Input**: Batches $\{\mathcal{B}_i\}_{i=1}^K$; trained reward function $\{\hat{R}_i\}_{i=1}^K$; randomly initialized $Q_D$, $G_D$ and $\xi_D$ jointly parameterized by $\theta$; task inference module $q_\phi$ with randomly initialized $\phi$; context set size $N_{\text{context}}$; training batch size $N$; triplet margin $m$

1: Learn single task policy $Q_i$, $G_i$, and $\xi_i$ from each data batch $\mathcal{B}_i$ using BCQ, $\forall i$
2: **repeat**
3:     Sample context set $\mathbf{c}_i$ from $\mathcal{B}_i$, $\forall i$
4:     **for** $i = 1, \ldots, K$ **do**
5:         Obtain the relabelled context set $\mathbf{c}_{j \rightarrow i}$ from $\mathbf{c}_j$ with $\hat{R}_i$ according to Alg. 3, $\forall j \neq i$
6:         Sample a subset of relabelled context set $\tilde{\mathbf{c}}_i$: $\tilde{\mathbf{c}}_i \sim \cup_j \mathbf{c}_{j \rightarrow i}$, $|\tilde{\mathbf{c}}_i| = N_{\text{context}}$
7:         Denote transitions in $\tilde{\mathbf{c}}_i$ originated from $\mathbf{c}_j$ as $\mathbf{x}_{j \rightarrow i}$
8:         Denote transitions in $\mathbf{x}_{j \rightarrow i}$ before relabelling as $\mathbf{x}_j$, $\mathbf{x}_j \in \mathbf{c}_j$
9:         Sample a subset $\mathbf{c}_{i,j}$ from $\mathbf{c}_i$ with $|\mathbf{c}_i| = |\mathbf{x}_j|$, $\forall j \neq i$
10:       Calculate the triplet loss $\mathcal{L}_{triplet}^i$

$$\mathcal{L}_{triplet}^i = \frac{1}{K-1} \sum_{j=1, j \neq i}^K \left[ d(q_\phi\left(\mathbf{x}_{j \rightarrow i}\right), q_\phi\left(\mathbf{c}_{i,j}\right)) - d(q_\phi\left(\mathbf{x}_{j \rightarrow i}\right), q_\phi\left(\mathbf{x}_j\right)) + m \right]_+ \quad (13)$$

11:       Combine $\mathbf{c}_i$ and $\tilde{\mathbf{c}}_i$ to form the new context set $\{\mathbf{c}_i, \tilde{\mathbf{c}}_i\}$
12:       Infer the posterior $q_\phi(\mathbf{z}|\{\mathbf{c}_i, \tilde{\mathbf{c}}_i\})$ over task identity from $\{\mathbf{c}_i, \tilde{\mathbf{c}}_i\}$
13:       Sample task identity $\mathbf{z}_i \sim q_\phi(\mathbf{z}|\{\mathbf{c}_i, \tilde{\mathbf{c}}_i\})$
14:       Sample training batch: $\{(s_t, a_t, r_t, s_t')\}_{t=1}^N$
15:       Calculate the value function distillation loss

$$\mathcal{L}_Q^i = \frac{1}{N} \sum_{t=1}^N \left[ Q_i(s_t, a_t) - Q_D(s_t, a_t, \mathbf{z}_i))^2 \right] + \beta \text{KL}(q_\phi(\mathbf{z}|\{\mathbf{c}_i, \tilde{\mathbf{c}}_i\})||\mathcal{N}(0,1)) \quad (14)$$

16:       Sample $N$ noises: $\nu_t \sim \mathcal{N}(0,1)$, $t = 1, \ldots, N$
17:       Obtain candidate action from $G_i$: $\hat{a}_t = G_i(s_t, \nu_t)$, $t = 1, \ldots, N$
18:       Calculate the candidate action generator distillation loss

$$\mathcal{L}_G^i = \frac{1}{N} \sum_{t=1}^N \left[ ||\hat{a}_t - G_D(s_t, \nu_t, \bar{\mathbf{z}}_i)||^2 \right] \quad (15)$$

19:       Calculate the perturbation generator distillation loss

$$\mathcal{L}_\xi^i = \frac{1}{N} \sum_{t=1}^N \left[ ||\xi_i(s_t, \hat{a}_t) - \xi_D(s_t, \hat{a}_t, \bar{\mathbf{z}}_i)||^2 \right] \quad (16)$$

20:     **end for**
21:     Calculate $\mathcal{L}_{triplet} = \frac{1}{K} \sum_{t=1}^K \mathcal{L}_{triplet}^i$
22:     Calculate $\mathcal{L}_Q = \frac{1}{K} \sum_{t=1}^K \mathcal{L}_Q^i$
23:     Calculate $\mathcal{L}_G = \frac{1}{K} \sum_{t=1}^K \mathcal{L}_G^i$
24:     Calculate $\mathcal{L}_\xi = \frac{1}{K} \sum_{t=1}^K \mathcal{L}_\xi^i$
25:     Update $\theta, \phi$ to minimize $\mathcal{L} = \mathcal{L}_{triplet} + \mathcal{L}_Q + \mathcal{L}_G + \mathcal{L}_\xi$
26: **until** Done

---

**Algorithm 5** Evaluation procedures of our model

---

**Input**: unseen task $\mathcal{M}$; learned multi-task policy

1: Initialize context set $\mathbf{c} \leftarrow \{\}$; initialize $q_\phi(\mathbf{z}|\mathbf{c}) = \mathcal{N}(0, 1)$
2: **repeat**
3:    Sample task identity $\mathbf{z} \sim q_\phi(\mathbf{z}|\mathbf{c})$.
4:    Collect one episode of transitions $\{(s_t, a_t, r_t, s'_t)\}_t$ from task $\mathcal{M}$ with multi-task policy conditioned on $\mathbf{z}$.
5:    Add $\{(s_t, a_t, r_t, s'_t)\}_t$ to $\mathbf{c}$.
6: **until** Done

**Output**: average episode returns, not counting the first two episodes

---

# F   Action selection and evaluation of the multi-task policy

$$\pi(s, \mathbf{z}) = \underset{a_m + \xi_D(s, a_m, \mathbf{z})}{\arg\max} \; Q_D\left(s, a_m + \xi_D\left(s, a_m, \mathbf{z}\right), \mathbf{z}\right) \qquad \nu_m \sim \mathcal{N}(0, 1)$$

Figure 10: Action selection. Given context set $\mathbf{c}$, $q_\phi$ infer the posterior over task identity, from which we sample the task identity $\mathbf{z}$. With the task identity $\mathbf{z}$, $G_D$ generates multiple candidate actions $a_m$ for state $s$. $\xi_D$ generates small corrections $\xi_D\left(s, a_m, \mathbf{z}\right)$ for the candidate actions $a_m$. The policy takes the corrected action $a_m + \xi_D\left(s, a_m, \mathbf{z}\right)$ with the highest value as estimated by $Q_D$.

In this section, we will describe the action selection procedures from the multi-task policy as shown in Fig. 10, and how we evaluate its performance.

Sampling action given a state from the multi-task policy is similar to the procedures of BCQ (Appendix B). The main difference is that the networks also take an inferred task identity $\mathbf{z}$ as input. Concretely, given a state $s$, the distilled candidate action generator $G_D$ generates multiple candidate actions $\{a_m = G_D(s, \nu_m, \mathbf{z})\}_m$ with random noise $\nu_m \sim \mathcal{N}(0, 1)$. The distilled perturbation generator $\xi_D$ generates a small correction term $\xi_D(s, a_m, \mathbf{z})$ for each state-candidate action pair. We take the corrected action with the highest value as estimated by the distilled value function $Q_D$. The action selection procedures can be summarized by:

$$\pi(s, \mathbf{z}) = \underset{a_m + \xi_D(s, a_m, \mathbf{z})}{\arg\max} \; Q_D\left(s, a_m + \xi_D\left(s, a_m, \mathbf{z}\right), \mathbf{z}\right), \; \{a_m = G_D\left(s, \nu_m, \mathbf{z}\right)\}_m, \; \nu_m \sim \mathcal{N}(0, 1). \tag{17}$$

We elaborate the evaluation procedures in Alg. 5. When testing on a new task, we do not have the ground truth task identity or any transition from the task to infer the task identity. We thus sample the initial task identity from the standard Gaussian prior in line 1. The task identity is kept fixed for the duration of the first episode. Afterwards, we use the collected transitions to infer the posterior and sample new task identities before each new episode as described in line 3. When calculating the average episode returns, we do not count the first two episodes' returns as what is done in [11].

# G   On Modifying the original HumanoidDir task distribution

We are concerned the original HumanoidDir task distribution is not suitable as a benchmark for multi-task RL because a policy trained from a single task can already obtain the optimal performance on unseen tasks. In particular, we train BCQ with transitions from one task and it obtains a similar return, as measured on unseen tasks ($993 \pm 33$), to SAC trained from scratch separately for each task ($988 \pm 19$).

In the HumanoidDir task distribution, each task is defined by a target running direction. The intended task is for the agent to run with maximal velocity in the target direction. The reward of each task can be defined as below:

$$R(s, a, s') = \text{alive\_bonus} + \alpha * \text{achieved\_velocity} \cdot \text{target\_direction}$$
$$- \text{quad\_ctrl\_cost} - \text{quad\_impact\_cost}, \tag{18}$$

where $\cdot$ denotes the inner product. Note that the two cost terms tend to be very small thus it will be reasonable to omit them in analysis. The alive\_bonus is the same across different tasks and is a constant. The target\_direction is different across tasks. $\alpha$ weights their relative contribution to the reward. If $\alpha$ is too small, the reward is dominated by the constant alive\_bonus. In this case, to achieve good performance, the agent does not need to perform the intended task. In other word, the agent does not need to infer the task identity to obtain good performance and only needs to remain close to the initial state while avoiding terminal states to maximize the episode length.

Prior works that use HumanoidDir set alive\_bonus $= 5.0$ and $\alpha = 0.25$. With such a small value for the reward coefficient $\alpha$, the reward is dominated by the alive\_bonus. We provide video to illustrate that in different tasks, the SAC-trained single-task policies display similar behaviors even though the different tasks have different running directions[4]. In most tasks, the SAC-trained policy controls the Humanoid to stay upright near the initial state, which is enough to obtain high performance. If a single policy that controls the agent to stay upright can achieve high performance in all tasks sampled from this task distribution, we argue that the learned multi-task policy in this task distribution can achieve near-optimal performance across tasks without the need to perform accurate task inference. In other word, this task distribution is not suitable to demonstrate the test-time task inference challenge identified in our work.

Therefore, we set $\alpha = 1.25$, which is the value used in the OpenAI implementation of Humanoid[5], and denote the modified task distribution as HumanoidDir-M. As is shown in the video, the SAC-trained agent in our case runs with significant velocity in the target direction. The optimal behaviors among the different tasks are thus sufficiently different such that the multi-task policy needs to infer the task identity to obtain high performance.

# H Details of the environmental settings and baseline algorithms

In this section, we will first provide the details of environmental settings in Appendix H.1, and then describe the baseline algorithms we compare against in Sec 5. We explain PEARL in Appendix H.2 and Contextual BCQ in Appendix H.3.

## H.1 Environment setups

We construct the task distribution UmazeGoal-M by modifying the maze-umaze-dense-v1 from D4RL. We always reset the agent from the medium of the U shape maze, while the goal locations is randomly initialized around the two corners of the maze.

The episode length is 1000 for HalfCheetahVel, which is the episode length commonly used when model-free algorithms are tested in the single-task variant of these task distributions. We use the same episode length 300 as D4RL for UmazeGoal-M. In the remaining task distributions, we set the episode length to be 200 due to constrained computational budget.

Table 9 provides details on each task distribution, including the number of training tasks and number of testing tasks. Note that the set of training tasks and the set of testing tasks do not overlap. The column "Interactions" specifies the number of transitions available for each task. With the selected number of interactions with the environment, we expect the final performance of training SAC in each task to be slightly below the optimal performance. In other word, we do not expect the batch data to contain a large amount of trajectories with high episode returns.

|  | Num train tasks | Num test tasks | Interactions | SAC returns | BCQ returns |
|---|---|---|---|---|---|
| HalfCheetahVel | 10 | 8 | 60K | $-121.3_{\pm 35.3}$ | $-142.7_{\pm 29.9}$ |
| AntDir | 10 | 8 | 200K | $920.9_{\pm 85.4}$ | $956.6_{\pm 83.8}$ |
| AntGoal | 10 | 8 | 300K | $-99.6_{\pm 33.9}$ | $-127.8_{\pm 36.7}$ |
| WalkerParam | 30 | 8 | 300K | $671.1_{\pm 106.4}$ | $692.6_{\pm 97.0}$ |
| HumanoidDir-M | 10 | 8 | 600K | $2116.1_{\pm 388.6}$ | $2190.9_{\pm 370.9}$ |
| UmazeGoal-M | 10 | 8 | 30K | $252.8_{\pm 6.5}$ | $258.3_{\pm 9.1}$ |

Table 9: Details of the experimental settings

## H.2 PEARL under Batch RL setting

Our works are very much inspired by PEARL [11], which is the state-of-the-art algorithm designed for optimizing the multi-task objective in various MuJoCo benchmarks. By including the results for PEARL, we demonstrate that conventional algorithms that require interaction with the environment during training does not perform well in the Multi-task Batch RL setting, which motivates our work.

To help readers understand the changes we made to adapt PEARL to the Batch RL setting, we reuse the notations from the original PEARL paper in this section. Detailed training procedures are provided in Algorithm 6. Without the privilege to interact with the environment, PEARL proceeds to sample the context set $c^i$ from the task batch $\mathcal{B}_i$ in line 5. The task inference module $q_\phi$, parameterized by $\phi$ takes as input the context set $c^i$ to infer the posterior $q_\phi(z|c^i)$. In line 6, we sample the task identity $z_i$ from $q_\phi(z|c^i)$. In line 7-9, the task identity $z_i$ combined with the RL mini-batch $b^i$ is further input into the SAC module. For task $i$, $\mathcal{L}^i_{actor}$ defines the actor loss, and $\mathcal{L}^i_{critic}$ defines the critic loss. $\mathcal{L}^i_{KL}$ constrains the inferred posterior $q(z|c^i)$ over task identity from context set $c^i$ to stay close to the prior $r(z)$. As shown in line 11, gradients from minimizing both $\mathcal{L}^i_{critic}$ and $\mathcal{L}^i_{KL}$ are used to train the task inference module $q_\phi$. We refer the readers to the PEARL paper for detailed definitions of these loss functions.

In PEARL, the context set is sampled from a replay buffer of recently collected data, while the training RL mini-batches (referred to as the RL batches in PEARL) is sampled uniformly from the replay buffer. This is not possible in the Multi-task Batch RL setting since all transitions are collected prior to training and are ordered arbitrarily. There is thus not a well-defined notion of "recently collected data".

---

**Algorithm 6** PEARL under Multi-task Batch RL setting (modified from Algorithm 1 in PEARL)

---

1: **Require:** Batches $\{\mathcal{B}_i\}_{i=1}^K$, learning rates $\alpha_1, \alpha_2, \alpha_3$
2: **while** not done **do**
3:     **for** step in training steps **do**
4:         **for** $i = 1, \ldots, K$ **do**
5:             Sample context $\mathbf{c}^i \sim \mathcal{B}^i$ and RL batch $b^i \sim \mathcal{B}^i$
6:             Sample $\mathbf{z} \sim q_\phi(\mathbf{z}|\mathbf{c}^i)$
7:             $\mathcal{L}_{actor}^i = \mathcal{L}_{actor}(b^i, \mathbf{z})$
8:             $\mathcal{L}_{critic}^i = \mathcal{L}_{critic}(b^i, \mathbf{z})$
9:             $\mathcal{L}_{KL}^i = \beta D_{\text{KL}}(q(\mathbf{z}|\mathbf{c}^i)||r(\mathbf{z}))$
10:         **end for**
11:         $\phi \leftarrow \phi - \alpha_1 \nabla_\phi \sum_i \left( \mathcal{L}_{critic}^i + \mathcal{L}_{KL}^i \right)$
12:         $\theta_\pi \leftarrow \theta_\pi - \alpha_2 \nabla_\theta \sum_i \mathcal{L}_{actor}^i$
13:         $\theta_Q \leftarrow \theta_Q - \alpha_3 \nabla_\theta \sum_i \mathcal{L}_{critic}^i$
14:     **end for**
15: **end while**

---

---

**Algorithm 7** Contextual BCQ (modified from Algorithm 1 in BCQ [9])

---

1: **Input:** Batches $\{\mathcal{B}_m\}_{m=1}^K$, horizon $T$, target network update rate $\tau$, mini-batch size $N$, max perturbation $\Phi$, number of sampled actions $n$, minimum weighting $\lambda$.
2: Initialize task inference module $q_\psi$ Q-networks $Q_{\theta_1}, Q_{\theta_2}$, perturbation network $\xi_\phi$, and VAE $G_\omega = \{E_{\omega_1}, D_{\omega_2}\}$, with random parameters $\psi$, $\theta_1$, $\theta_2$, $\phi$, $\omega$, and target networks $Q_{\theta_1'}, Q_{\theta_2'}, \xi_{\phi'}$ with $\theta_1' \leftarrow \theta_1, \theta_2' \leftarrow \theta_2, \phi' \leftarrow \phi$.
3: **repeat**
4:     **for** $m = 1, \ldots, K$ **do**
5:         Sample $N$ transitions $\{(s, a, r, s')_t\}_t$ from each $\mathcal{B}_m$
6:         Sample context set $\mathbf{c}_m$ from $\mathcal{B}_m$
7:         Sample task identity $\mathbf{z}_m$ from the inferred posterior $q_\psi(\mathbf{z}|\mathbf{c}_m)$
8:         $\mu, \sigma = E_{\omega_1}(s, a, \mathbf{z}_m), \quad \tilde{a} = D_{\omega_2}(s, \nu, \mathbf{z}_m), \quad \nu \sim \mathcal{N}(\mu, \sigma)$
9:         $\omega \leftarrow \text{argmin}_\omega \sum(a - \tilde{a})^2 + D_{\text{KL}}(\mathcal{N}(\mu, \sigma)||\mathcal{N}(0, 1))$
10:         Sample $n$ actions: $\{a_i \sim G_\omega(s', \mathbf{z}_m)\}_{i=1}^n$
11:         Set value target $y$ (Eq. 20)
12:         $\theta \leftarrow \text{argmin}_\theta \sum(y - Q_\theta(s, a, \mathbf{z}_m))^2$
13:         $\phi \leftarrow \text{argmax}_\phi \sum Q_{\theta_1}(s, a + \xi_\phi(s, a, \mathbf{z}_m, \Phi), \mathbf{z}_m), a \sim G_\omega(s, \mathbf{z}_m)$
14:         $\psi \leftarrow \text{argmin}_\psi \sum(y - Q_\theta(s, a, \mathbf{z}_m))^2$
15:         Update target networks: $\theta_i' \leftarrow \tau\theta + (1 - \tau)\theta_i'$
16:         $\phi' \leftarrow \tau\phi + (1 - \tau)\phi'$
17:     **end for**
18: **until** iterates for $T$ times

---

## H.3 Contextual BCQ

We reuse the notations from the original BCQ paper [9] to help reader understand how we modify modify BCQ to train multi-task policy by incorporating a task inference module. We refer to this method as Contextual BCQ and use it to serve as our baseline methods. By comparing this baseline, we argue that the problem we are facing cannot be solved by simply combining the current Batch RL algorithm with a simple task inference module. Next, we will start by providing a brief introduction of the training procedures of BCQ.

Batch Constrained Q-Learning (BCQ) is a Batch RL algorithm that learns the policy from a fixed data batch without further interaction with the environment [9]. By identifying the extrapolation error, BCQ restricts the action selection to be close actions taken in batch. Specifically, it trains a conditional variational auto-encoder $G$ [19] to generate candidate actions that stay close to the batch for each state $s$. A perturbation model $\xi$ will generate a small additional correction term to induce limited exploration for each candidate action in the range $[-\Phi, \Phi]$. The perturbed action with the highest state action value as estimated by a learned value function $Q$ will be selected.

By modifying BCQ to incorporate incorporate module detail, the training procedures of Contexual BCQ can be detailed in Alg. 7. As the original BCQ algorithm, we maintain two separate Q function networks $Q_{\theta_1}, Q_{\theta_2}$ parameterized by $\theta_1, \theta_2$, a generative model $G_\omega = \{E_{\omega_1}, D_{\omega_2}\}$ parameterized by $\omega$, where $E_{\omega_1}, D_{\omega_2}$ are the encoder and decoder, and a perturbation generator $\xi_\phi$ parameterized by $\phi$. For the $Q_{\theta_1}, Q_{\theta_2}$ and $\xi_\phi$, we also maintain their corresponding target networks $Q_{\theta_1'}, Q_{\theta_2'}$ and $\xi_{\phi'}$. Compared with the original BCQ, all these networks will take in the inferred task identity $\mathbf{z}$ as an extra input, which is generated by the task inference module as $q_\psi$ parameterized by $\psi$.

We use $m$ to index the task. From each task batch $\mathcal{B}_m$, in line 5, we sample a context set $\mathbf{c}_m$ and $N$ transitions $\{s_t, a_t, r_t, s_t'\}_t$, where $t$ indexes the transition. For simplicity, we denote the transitions with the shorthanded $\{(s, a, r, s')_t\}_t$. In line 7, $q_\psi$ takes as input the context set $\mathbf{c}_m$ and infer a posterior $q_\psi(\mathbf{z}|\mathbf{c}_m)$ over the task identity, from which we sample a task identity $\mathbf{z}_m$.

Line 8-9 provide the procedures to train the generative model $G_\omega = \{E_{\omega_1}, D_{\omega_2}\}$. Specifically, $E_{\omega_1}$ takes as the input the state-action pair $(s, a)$ and task identity $\mathbf{z}_m$ and output the mean $\mu$ and variance $\sigma$ of a Gaussian distribution $\mathcal{N}(\mu, \sigma)$. That is, $\mu, \sigma = E_{\omega_1}(s, a, \mathbf{z}_m)$. From $\mathcal{N}(\mu, \sigma)$, we sample a noise $\nu$ and input it to the decoder $D_{\omega_2}$ together with $s$ and $\mathbf{z}_m$ to obtain the reconstructed action $\tilde{a} = D_{\omega_2}(s, \nu, \mathbf{z}_m)$. We train $G_\omega$ by minimizing

$$\sum (a - \tilde{a})^2 + D_{\text{KL}}(\mathcal{N}(\mu, \sigma) || \mathcal{N}(0, 1)). \tag{19}$$

Line 10-12 provide the procedures to train the Q value functions. For each next state $s'$ in the training batch, we can obtain $n$ candidate actions $\{a_i \sim G_\omega(s', \mathbf{z}_m)\}_{i=1}^n$ from the generative model $G_\omega$. This is done by sampling $n$ noises from the prior $\mathcal{N}(0, 1)$ and input to decoder $D_{\omega_2}$ together with $s'$, as shown in line 10. For each of the candidate action $a_i$, the perturbation model $\xi_\phi$ will generate a small correction term $\xi_\phi(s', a_i, \mathbf{z}_m, \Phi) \in [-\Phi, \Phi]$. We denote the perturbed actions as $\{a_i = a_i + \xi_\phi(\hat{s}', a_i, \mathbf{z}_m, \Phi)\}_{i=1}^n$. Therefore, the learning target for both of the Q function network is given by

$$y = r + \gamma \max_{a_i} \left[ \lambda \min_{j=1,2} Q_{\theta_j'}(s', a_i, \mathbf{z}_m) + (1 - \lambda) \max_{j=1,2} Q_{\theta_j'}(s', a_i, \mathbf{z}_m) \right], \tag{20}$$

where $a_i$ is selected from the set of perturbed actions and the minimum weighting $\lambda$ can be set to control the overestimation bias and future uncertainty. We also use Eq. 20 to train the task inference module $q_\psi$ in line 14.

In line 13, $\xi_\phi$ is trained to generate a small perturbation term in range $[-\Phi, \Phi]$ so that the perturbed candidate actions $a + \xi_\phi(s, a, \mathbf{z}_m, \Phi)$ can maximize the state action value estimated by the Q function. Note that the candidate actions $a$ are output by the generative model $G_\omega$. The loss function to train $\xi_\phi$ thus can be formulated as

$$\sum Q_{\theta_1}(s, a + \xi_\phi(s, a, \mathbf{z}_m, \Phi), \mathbf{z}_m), \quad a \sim G_\omega(s, \mathbf{z}_m) \tag{21}$$

Figure 11: Ablation study of the triplet margin on four task distributions. The horizontal axis indicates the number of epochs. The vertical axis indicates the average episode return. The shaded areas denote one standard deviation.

## I  Additional experimental results

In addition to the results already presented in Sec. 5.2, this section presents more experimental results to further understand different design choices of our model. We evaluate the performance of our model when we set different values of the *triplet margin* in Appendix I.1. We present the ablation study of the reward ensemble in Appendix I.2.

### I.1  Ablation study of the triplet margin

Recall the triplet loss for task $i$ defined in Sec. 3.2,

$$\mathcal{L}_{triplet}^{i} = \frac{1}{K-1} \sum_{j=1, j \neq i}^{K} \left[ \underbrace{d\big(q_\phi\left(\mathbf{c}_{j \to i}\right), q_\phi\left(\mathbf{c}_i\right)\big)}_{\substack{\text{Ensure } \mathbf{c}_{j \to i} \text{ and } \mathbf{c}_i \text{ infer} \\ \textit{similar} \text{ task identities}}} - \underbrace{d\big(q_\phi\left(\mathbf{c}_{j \to i}\right), q_\phi\left(\mathbf{c}_j\right)\big)}_{\substack{\text{Ensure } \mathbf{c}_{j \to i} \text{ and } \mathbf{c}_j \text{ infer} \\ \textit{different} \text{ task identities}}} + m \right]_{+},$$

where $\mathbf{c}_{j \to i}$ denotes the set of transitions relabelled by the reward function $\hat{R}_i$ of task $i$, and $\mathbf{c}_i$ denotes the context set for task $i$. We include a positive term $m$ referred to as *triplet margin* when calculating the triplet loss. With this term, we expect that $d\big(q_\phi\left(\mathbf{c}_{j \to i}\right), q_\phi\left(\mathbf{c}_i\right)\big)$ is at least smaller than $d\big(q_\phi\left(\mathbf{c}_{j \to i}\right), q_\phi\left(\mathbf{c}_j\right)\big)$ by $m$.

Here, we examine how the performance of our algorithm changes when varying the value of the triplet margin $m$. Specifically, we set $m = 0.0, 2.0, 4.0, 8.0$ and show the results on the five task distributions. As can be seen in Figure 11, the performance of our model is in-sensitive to the value of the triplet loss margin.

Figure 12: Results on HumanoidDir-M. When we use the multi-task policy trained without a reward ensemble to initialize SAC, performance has higher variance and converges to a lower value compared to using the ensemble. The horizontal axis indicates the number of training epochs. The vertical axis indicates the average episode return. The shaded areas denote one std.

## I.2 Ablation study of reward prediction ensemble

In Sec. D, we describe the use of a reward ensemble to increase reward prediction accuracy when relabelling transitions. In this section, we demonstrate the benefit of the reward ensemble. Recall that in Sec. 5.3, we use the trained multi-task policy as an initialization when further training is allowed on the unseen tasks. Using the reward ensemble when training the multi-task policy leads to higher performance when the trained multi-task policy is used as an initialization. Training our model without using a reward ensemble means we use one instead of an ensemble of networks to approximate the reward function for each training task. As shown in Figure 12, if the multi-task policy is trained without using the reward ensemble, when the multi-task policy is used as an initialization, the performance has high variance and has smaller asymptotic value.

**Algorithm 8** Imitation procedures

**Input**: Unseen testing task $\mathcal{M}$; trained multi-task policy; randomly initialized single-task policy $\pi_\psi$;
1: Sample 10K transitions $\mathcal{R} = \{(s_t, a_t, r_t, s'_t)\}_t$ from $\mathcal{M}$ using the multi-task policy and infer the task identity $\mathbf{z}$.
2: **while** not done **do**
3:      Sample a transitions $(s, a, r, s')$ from $\mathcal{R}$
4:      Obtain $\mu_s, \Sigma_s = \pi_\psi(s)$
5:      Calculate the log likelihood $\mathcal{J} = -\frac{1}{2} \log |\Sigma_s| - \frac{1}{2} (a - \mu_s)^T \Sigma_s^{-1} (a - \mu_s) + constant$
6:      $\psi \leftarrow \psi + \nabla_\psi \mathcal{J}$
7: **end while**
**Output**: Initialized single-task policy $\pi_\psi$; inferred the task identity $\mathbf{z}$

## J   Details of using multi-task policy to initialize SAC

In this section, we provide more detailed illustrations of using the learned multi-task policy to initialize training on unseen tasks (Sec. 5.3). We provide the pseudo-code for the SAC initialized by our methods in Appendix J.1. To demonstrate that the acceleration of convergence is really thanks to the transferability of our multi-task policy, we compare it against the variation of SAC, where we train the identically initialized two Q functions with different different mini-batch sampled from the replay buffer and also maintain a target policy network to stabilize the training process.

### J.1   Pseudo-code for SAC initialized by our methods

To help readers understand the changes we made, we reuse the notations from the original SAC paper [40] in this section. We first provide the pseudo-codes for initializing the new single-task policy $\pi_\psi$ parameterized by $\psi$ via imitation learning procedures in Alg. 8. The whole training procedures are detailed in Alg. 9 by modifying pseudo-code provided in [64].

As is commonly done, the policy $\pi_\psi$ outputs the mean $\mu_s$ and $\Sigma_s$ of a Gaussian distribution for each state $s$, which characterizes the pdf of action selection, i.e. $p(a|s) \sim \mathcal{N}(\mu_s, \Sigma_s)$). In line 1 of Alg. 8, we first collect 10K transitions using the multi-task policy in the unseen testing task $\mathcal{M}$. In line 2-7, We train $\pi_\psi$ to maximize the log likelihood of the action selections inside the collected data. Note that we also infer a task identity $\mathbf{z}$ from the 10K transitions in line 1. Its usage will be illustrated next.

With the initialized single-task policy $\pi_\psi$ and inferred the task identity $\mathbf{z}$, we now turn to elaborate the whole process of the SAC initialized by the learnt multi-task policy. We detail the training procedures in Alg. 9. Compared with the standard SAC, in line 1 we initialize both the two Q function networks $Q_{\theta_1}$ and $Q_{\theta_2}$ identically with $Q_D$. However, in line 11-15, we train them using different batch data $B_1$ and $B_2$ sampled from the replay buffer $\mathcal{D}$ to stabilize the training process [44]. In addition to maintain target networks $Q_{\theta_{\text{targ},1}}$ and $Q_{\theta_{\text{targ},2}}$ for each Q function, we also maintain a target policy network $\pi_{\psi_{\text{targ}}}$.

Note that we perform imitation learning to initialize a new single-task policy instead of using the candidate action and perturbation generator to directly initialize the policy as the Q value function. To directly transfer the parameters of the candidate action and perturbation generator, we can initialize the action selection module with the distilled candidate action and perturbation generator and train them to generate action $a$ for state $s$ that maximizes the expected $Q(s, a)$ over a training batch. However, we find that this training procedures converge to a lower asymptotic performance.

**Algorithm 9** SAC initialized by our method

---

**Input**: unseen testing task $\mathcal{M}$; initialized single-task policy $\pi_\psi$; inferred the task identity $\mathbf{z}$; Q functions $Q_{\theta_1}, Q_{\theta_2}$ both initialized by $Q_D$ (from the multi-task policy); empty replay buffer $\mathcal{D}$

1: Set target parameters equal to main parameters $\theta_{\text{targ},1} \leftarrow \theta_1$, $\theta_{\text{targ},2} \leftarrow \theta_2$, $\psi_{\text{targ}} \leftarrow \psi$
2: **repeat**
3:     Observe state $s$ and select action $a \sim \pi_\psi(\cdot|s)$
4:     Execute $a$ in task $\mathcal{M}$
5:     Observe next state $s'$, reward $r$, and done signal $d$ to indicate whether $s'$ is terminal
6:     Store $(s, a, r, s', d)$ in replay buffer $\mathcal{D}$
7:     If $s'$ is terminal, reset environment state.
8:     **if** it's time to update **then**
9:         **for** $j$ in range(however many updates) **do**
10:             **for** $i = 1, 2$ **do**
11:                 Randomly sample a batch of transitions, $B_i = \{(s, a, r, s', d)\}$ from $\mathcal{D}$
12:                 Compute targets for the Q functions:

$$y(r, s', d) = r + \gamma(1 - d)\left(\min_{i=1,2} Q_{\theta_{\text{targ},i}}(s', \tilde{a}', \mathbf{z}) - \alpha \log \pi_\psi(\tilde{a}'|s')\right), \quad \tilde{a}' \sim \pi_\psi(\cdot|s')$$

13:                 Update Q-functions $i$ by one step of gradient descent using

$$\nabla_{\theta_i} \frac{1}{|B_i|} \sum_{(s,a,r,s',d)\in B_i} \left(Q_{\theta_i}(s, a, \mathbf{z}) - y(r, s', d)\right)^2$$

14:             **end for**
15:             Using the transitions from $B_2$, update policy by one step of gradient ascent using

$$\nabla_\psi \frac{1}{|B_2|} \sum_{s\in B_2} \left(\min_{i=1,2} Q_{\theta_i}(s, \tilde{a}_\psi(s), \mathbf{z}) - \alpha \log \pi_\psi\left(\tilde{a}_\psi(s)|s\right)\right),$$

    where $\tilde{a}_\psi(s)$ is a sample from $\pi_\psi(\cdot|s)$ which is differentiable wrt $\psi$ via the reparametrization trick [19].
16:             Update target networks with

$$\theta_{\text{targ},i} \leftarrow \rho\theta_{\text{targ},i} + (1 - \rho)\theta_i \qquad\qquad \text{for } i = 1, 2$$
$$\psi_{\text{targ}} \leftarrow \tau\psi_{\text{targ}} + (1 - \tau)\psi$$

17:         **end for**
18:     **end if**
19: **until** convergence

---

— : SAC initialized by our multi-task policy     — : A variation of SAC, randomly initialized

Figure 13: Comparison between the SAC initialized by our method and a variation of SAC. This variation trains two identically initialized Q functions with different mini-batches sampled from the replay buffer. Moreover, it also maintains a target policy network. The two methods share the same network sizes and architecture across all settings. In the figures above, the horizontal axis indicates the number of environment steps. The vertical axis indicates the average episode return. The shaded areas denote one standard deviation.

## J.2   Comparison with a variation of SAC

In Sec. 5.3, we show that SAC initialized by our method can significantly speed up the training on unseen testing tasks. An astute reader will notice that our implementation of Soft Actor Critic in Appendix J.1 is different from the reference implementation. We maintain two identically initialized value functions by training them using different mini-batches. In the reference implementation, the two value functions are initialized differently but trained using the same mini-batch.

To ensure the performance gain is really thanks to the good initialization provided by the multi-task policy, we compare its performance with a variation of SAC. Specifically, we initialize the two Q value functions identically but train them with different mini-batches sampled from the replay buffer. Moreover, we also maintain a target policy network as what is done in line 17 of Alg. 9. As shown in Figure 13, we can still observe that the SAC initialized by our methods outperform this variation of SAC. The unseen tasks used for evaluating the variation of SAC are the same as those used for testing the SAC initialized by our methods. The two methods share the same network sizes and architecture across all settings.

# K   Computing infrastructure and average run-time

Our experiments are conducted on a machine with 2 GPUs and 8 CPUs. Table 10 provides the runtime for each of the experiment on all the task distributions.

|                          | AntDir | AntGoal | HDir | UGoal | HalfCheetahVel | WalkerParam |
|--------------------------|--------|---------|------|-------|----------------|-------------|
| Our full model           | 4.5    | 4.6     | 5.3  | 4.0   | 26.6           | 21.4        |
| Contextual BCQ           | 5.9    | 5.9     | 5.3  | 5.5   | 17.2           | 13.1        |
| PEARL                    | 8.2    | 7.6     | 8.1  | 6.9   | 16.5           | 11.1        |
| No transition relabelling| 4.5    | 4.6     | 4.6  | 4.6   | 17.1           | 6.4         |
| No triplet loss          | 4.4    | 5.6     | 4.1  | 4.6   | 17.4           | 8.2         |
| Neither                  | 4.5    | 4.6     | 3.9  | 6.2   | 16.7           | 7.5         |
| SAC init by our method   | 4.2    | 4.3     | 5.3  | -     | 0.6            | 4.2         |

Table 10: Runtime of each experiment. The unit is hours. When calculating the runtime for algorithms that learn a multi-task policy, we exclude neither the time to generate the task buffers nor the time to learn single-task BCQ policies. The abbreviation HDir and UGoal stands for HumanoidDir-M and UmazeGoal-M, respectively. The runtime for SAC initialized by our methods is calculated by average across tasks from the corresponding task distribution.

## Footnotes

[4]Videos are provided: `https://www.youtube.com/channel/UCWrYNNRgZzqxnhfOYbgNmkA`

[5]OpenAI implementation of Humanoid-v2 is provided here `https://github.com/openai/gym/blob/master/gym/envs/mujoco/humanoid.py`