[Reviews · NeurIPS 2020]

Review 1

Summary and Contributions: The paper proposes an interesting take to perform multi-task reinforcement learning with offline data. The paper proposes utilizes BCQ and a relatively novel triplet loss to perform better task-id in the setting. The paper is clear and thorough in the writing, and the empirical evaluation conducted is properly done over 2 different baselines.

Strengths: - The problem considered is well motivated and clearly explained - The paper tells a nice and clear story, which flows well and is well motivated. - The paper shows clearly significant improvements over the considered baseline on 5 of the 6 tasks. - The paper also is well-written and easy to follow. - The ablation also adds an import study to the paper.

Weaknesses: - The paper lacks a very significant technical contribution, although, I believe that the paper is has sufficient technical contribution as it stands. The key novelty in the paper is the experimental protocol as well as the triplet loss, since distillation is a commonly used technique. Please correct me if I am mistaken in my belief however. - The baselines selected are somewhat poor. Lots of methods have since outperformed PEARL, most recently Meta-GenRL [1] which appears to be the current SOTA in similar line of study. Contextual BCQ also seems to be an ablation more than a baseline, I could be wrong. - The related works section is very short and rushed. The paper exists in a place between several fields (namely Meta RL, MTL and Batch RL), so I would like to see a more thorough analysis on what the differences are between the literature that exists. - NIT: Fig 1, 4 and 5 is very small and difficult to read [1] https://arxiv.org/abs/1910.04098

Correctness: Yes, the paper runs important ablation analysis, and tests over 6 environments while making appropriate changes to baselines (PEARL) to ensure they "work". The paper claimes: "To the best of our knowledge, we are the first to highlight the issue of the task inference module learning the wrong correlation from biased dataset" which I do believe to be true.

Clarity: Very well written. I would prefer the related works be significantly expanded highlighting the differences (ie why do you suspect CQL [1] or BEAR [2] or other Batch RL papers to fail in this context?). I would also like all the figures to be significantly larger in size, than their current size. [1] https://arxiv.org/abs/2006.04779 [2] https://arxiv.org/abs/1906.00949

Relation to Prior Work: See above. The paper needs to do a significantly better job of highlighting the differences between the fields it borrows ideas from. I would also appreciate a small discussion on triplet loss / metric learning, why margin loss, compared to the many different triplet loss schemes that exist. [1] is a good resource for the topic. [1] https://arxiv.org/abs/2002.08473

Reproducibility: Yes

Additional Feedback: Happy to raise my score if my comments are addressed. I think this is a good paper which should be accepted if most importantly the connection and relation to prior work is made more clear. Adding a baseline of Meta-GenRL might also significantly improve the paper in my eyes. POST REBUTTAL: I have read the rebuttal, and the other reviews. One of the reviewers raised valid improvements to the submission, although I disagree with some others. But in light of that, I have decided to stick to the same score as pre-rebuttal.


Review 2

Summary and Contributions: The paper tackles the multi-task batch reinforcement learning problem. Given the datasets collected from different tasks, how to train a policy that perform well in unseen tasks? The proposed method infers the task identity from the collected data via modelling the dependence on states, actions, and rewards. Subsequently, the proposed method learns a task conditioned policy to complete the downstream tasks. To improve the task inference, the authors proposed a novel utilisation of a triplet loss. In the experiments, the authors show that the trained policy as an initialisation leads to faster convergence compared to randomly initialisations.

Strengths: The paper addresses a novel topic: multi-task batch reinforcement learning, which emerges from the previous multi-task and batch reinforcement learning works. The paper proposes a triplet loss function for robust task inference and evaluate them in Mujoco environments. The paper addresses the issue of the task inference module learning the wrong correlation from biased dataset. The proposed method includes a novel triplet loss and the approximation of reward function and the use of relabelling for mining hard negative examples.

Weaknesses: The novelty of the paper is relative weak given the previous work on multi-goal and multi-task RL, batch RL, and Meta RL. The work is based on Behaviour Constrained Q-learning (BCQ), which demonstrates good performance in single-task batch reinforcement learning. The proposed method first uses BCQ to learn different policy for each tasks, then, distill a single network by adding a tasks inference module. This module should recover the task identity and its corresponding skill set. And the task inference is done via the triple loss design. Overall, the author transfers BCQ to the multi-task domain. The experiments were carried out in the ant, halfcheeta, humanoid, and walker environments. There is a recently proposed offline reinforcement learning dataset, called D4RL. In the D4RL dataset, there are also multi-goal / task environments. It would be better if the author could also show the performance of the proposed method in the D4RL dataset in the future. Post-rebuttal: After reading the authors' rebuttal, I keep my score, which is positive. Thanks authors for adding some small additional experiments to address the concerns.

Correctness: Yes, the claims and the method are correct. The empirical methodology is correct.

Clarity: Yes, the paper is well written.

Relation to Prior Work: Yes.

Reproducibility: Yes

Additional Feedback:


Review 3

Summary and Contributions: This paper tackles multi-task batch reinforcement learning. The objective is to learn a policy on offline data collected by a reasonable agent, that performs well on unseen tasks from a similar distribution. This is both a task inference and policy learning problem. At test, we do not have access to the the task id or reward function a priori. This paper describes previous approaches to the problem and shows some drawbacks of more standard approaches. Specifically, if state-action pairs do not overlap between tasks (as these have been produced by a competent agent), the task inference model learns to ignore rewards to infer tasks. At test time this often leads to incorrect inference of task and thus a sub-optimal policy. The solution proposed is two fold: to relabel state transitions to create a distribution shift, forcing the agent to learn to pay attention to rewards functions, and a novel application of a triplet loss to enforce this. This is combined with bath constrained q-learning in standard ways.

Strengths: This paper is very well written -- it is concise, to the point and clearly explains and motivates the need for a new method and it's workings. There is a lot of scope for the learnings from this paper to be applied to other multi-task learning domains. They explain why this problem does not occur for eg in Atari where state spaces are unambiguous and test on a suite of Mujoco tasks. The discussion and ablations are both clear and the results promising.

Weaknesses: The paper could be improved by testing on a larger suite of tasks. These don't necessarily have to be more complex but for example a toy gridworld with different goals / reward functions would be able to motivate the problem more thoroughly. The related work section could also be expanded. While many papers in the literature are cited, the discussion around them could be expanded, providing more context to a reader new to the field.

Correctness: I haven't implemented the method to test correctness but it is clearly motivated and seems to me a novel and simple solution.

Clarity: This paper is very well written.

Relation to Prior Work: The contributions are clearly and concisely stated.

Reproducibility: Yes

Additional Feedback:


Review 4

Summary and Contributions: The authors identify, describe and address a problem occurring in the Multi-task Batch RL setting when test task identities have to be inferred from a given data. The paper argues that a given task should be inferred based on states, actions and also rewards (to generalize well), while the task inference modules tend to ignore rewards if not regularized. The authors find a cause which is a large divergence in state-action distribution for tasks. As a solution, the paper proposes to regularize an inference module with a data augmentation trick -- the ground truth rewards are replaced with rewards that would be given when the agent was tasked with another task. This additional data is leveraged only in a separate (added) loss used to train the inference module.

Strengths: The paper finds a surprising case in the Batch RL setting where machine learning models rely on shortcuts to solve their tasks and hence do not generalize as expected. The case is extensively analyzed and a solution aiming at the found cause is proposed. Some relevant ablations of the proposed method are implemented and compared.

Weaknesses: The main weakness of the method is a reliance on accurate relabelling. The paper argues that actor-critic networks got casually confused due to (almost) disjoint task distributions and then hopes that reward models will not have the same problem. However, it seems that the problem also affects reward models, as a reward ensemble is used in the experiments. There is no ablation study to investigate the necessity of this ensemble in the offline setting. * There is one somehow related experiment in the supplementary material but it is in an online setting (similarly to ablation study in 5.3), so it is not as informative as it can be. Can you explain why you did not use the setting from 5.1 and 5.2 to evaluate this component of your model? The experiment results are not convincing. The proposed method improves significantly on 2 out of 6 tasks. However, one task (HumanoidDir) is modified to tease out the difference. It is really misleading as the results are reported under the original name. I find it not fair to modify existing tasks to make the proposed method look better. I do understand the arguments for that (lines 203 - 207) but I think that results on both the original and the modified tasks should be reported. Hence, WalkerParam is the only original task where the proposed method works better. However, as shown in Figure 6, ablated models achieve similar results on this task. Hence, the improvement is NOT due to proposed improvements (relabelling and triplet loss). The ablation studies for other tasks suggest that the differences are not significant. All tasks are solved with relatively good scores by all methods. I would like to see an ablation study when the ground truth rewards for the other tasks are used for relabelling to see ‘the upper-bound’ of the method assuming the reward models generalize perfectly. Have you tried it? As this procedure may be practical in some cases, such results can strengthen the paper. For example, in manipulation tasks on the real robot, it is not rare to annotate the data with separately trained reward models for training tasks. Finally, it is not described why the extra relabelled data is not used to train other components (critic). Is it due to the low quality of these rewards? Has it been tested?

Correctness: As mentioned before, the paper assumes that tasks have significantly different state-action distributions (which is a valid assumption) and argues that this is the main culprit and motivation for the work. However, the same applies to reward model learning. It is discussed (lines 183 - 190), which is a big plus. However, I do not find it persuading. ‘Sophisticated’ architectures and structural causal models are proposed as a solution. But then, why do not apply them directly to policy training? I find the last argument (lines 188-190) valid but no ablation studies are presented to justify it (for example using a mix of ground truth rewards for the other tasks and random rewards).

Clarity: The description of the problem is very clear and intuitive. The method is also presented in a lucid way. However, I do not understand why so much is devoted to BCQ (lines 64 - 72, 107 - 116, and some other bits). For me, it appears that the proposed method is NOT specific to BCQ and can be in principle applied to any method that uses critic, or even more generally to ANY method when the task inference module can be successfully trained using triplet loss solely. Am I right here? If so, I think that it would be more clear to explain the core method separately, and then defer details on how we adapt the used algorithm (BCQ here) to be conditioned on the task? I think that the fact that the method can be applied with other batch RL methods is the advantage of the method and the current presentation masks it (and is harder to read). The sections about experiments should be improved. For example, the tasks should be explained in the main text, at least shortly. Now, the reader has to be familiar with the prior works (for example PEARL) to even understand what are subtasks in each environment (goal / direction etc.). Unfortunately the knowledge of the tasks is assumed even before the experimental section (e.g. the papers refers to ‘direction’ in line 131). More importantly, it is not mentioned nor in the paper, nor in the supplementary material, what are inputs for trained models. Is everything trained from low dimensional MuJoCo states? Is the position of the agent and the goal provided? If so, the task for reward models is extremely easy and then it may be the reason why the rewards can generalize better than the policy itself.

Relation to Prior Work: I think that the paper does a really good job here.

Reproducibility: No

Additional Feedback: AFTER REBUTTAL: The authors address a lot of my concerns in their response, including running experiment on the new environment, comparison to a model using ground truth rewards and an ablation study regarding reward prediction (though the last one does not compare final agent performance but only reward accuracy). I really appreciate the work and the new results. On the other hand, I think that there are still a lot of improvements needed (mainly these I mentioned in the clarity section) and they should be incorporated in the new version submitted to another venue. I raised my score but I am still against accepting the paper.

[Author Response · NeurIPS 2020]

We thank all reviewers for their feedback. Answers to reviews are denoted R2, R3, R5, R6.

R2: We feel our technical contribution is significant. Since offline data is essentially free
for many applications, RL methods should use it whenever possible. This is especially true
because practical deployments of RL are bottle-necked by its poor sample efficiency. In
particular, results in Sec. 5.3, where we use our policy to initialize an RL algorithm show
a substantial gain in performance, even in the complex HumanoidDir environment ($64\%$
improvement). As far as we know, we are the first to demonstrate such large gain, using only
offline data from other tasks and without knowledge of identity and reward of the test task.

R2: Concerning readability, we will increase figure size.

R2, R3, R5: We performed new experiments. In response to R2, MetaGenRL is designed
for online RL while we focus on batch (offline) RL. MetaGenRL relies on DDPG to learn
accurate value estimates, which are known to diverge in batch RL (as shown by the BCQ
paper). This means that MetaGenRL is not a strong baseline, as confirmed by our experiment
in Fig. b, where its performance quickly plummets and does not recover with more training
epochs. Combining MetaGenRL and our method would be interesting since MetaGenRL
generalizes to out-of-distribution tasks, but is beyond the scope of the paper. As suggested
by R3, we add results on D4RL. We didn't know about D4RL when writing the paper (it
is a recent preprint), but we ran the experiment on maze2d-umaze now (Fig. a). In this
experiment, we train with offline data and evaluate their performance without further training
on unseen navigation targets. Our model significantly outperforms the baselines and the
ablations. We will provide more analysis on this environment in the paper as R5 suggests.

R2, R3, R5: We are happy to extend the related work section and discuss all mentioned
papers. Regarding CQL and BEAR, they are single-task Batch RL algorithms and as such are
not directly applicable to multi-task Batch RL. We will discuss topics from the deep metric
learning paper: embedded samplers, the effect of mini-batch diversity and the correlation
between embedding space compression and generalization in RL. Also, since R3 mentions
novelty as a relative weakness, we would be grateful if R3 could provide us with more
references. The use of the triplet loss in this context is novel and opens up new research
directions to determine what the best metric learning loss for RL is.

R6: It is in fact possible to learn a good reward model. Existing model-based RL algorithms
necessarily rely on the ability to learn good reward models that generalize. Since the reward
function is a mapping from state-action pairs to scalar reward, it is in general much simpler
than the task identity function whose inputs are complex high dimensional sets and which maps to a high dimensional
embedding space. Moreover, unlike task inference, reward learning can be accomplished for each task independently.
Empirically, in Figure c, we show that our reward model indeed achieves low error on state-action pairs from another
task, both with and without an ensemble. Moreover, we did an ablation with the ground truth reward you suggested
on the D4RL maze2d-umaze environment (Fig. a). While using an oracle for the ground truth reward produces a
performance improvement, final performance is close to using our method with learned reward.

R6: We are concerned the original HumanoidDir environment is not suitable as a benchmark for multi-task RL because
a single-task policy already obtains high performance on unseen tasks. In particular, we train BCQ with transitions
from one task and it obtains a similar return, as measured on unseen tasks, ($993 \pm 33$) to SAC trained from scratch
separately for each task ($988 \pm 19$). You are right that we should have used a different name for the environment. We
will change the name and show results for both versions in the final version of the paper.

R6: Concerning performance, on AntDir, AntGoal and HumanoidDir, we outperform the best baseline Contextual BCQ
by $25\%$, $26\%$, $28\%$ in terms of mean return. On those 3 tasks, we outperform the best ablation no_transition_relabelling
by $20\%$, $26\%$, $14\%$. Our experiment on D4RL also shows clear improvement over baselines and ablations (Fig a.).
On WalkerParam, we agree with your analysis and will clarify in the paper that the performance improvement in
WalkerParam comes from distillation. We hypothesize that WalkerParam and HalfCheetahVel do not benefit from
reward relabelling because they are lower-dimensional, hence random sampling will lead to lower divergence in
state-action distribution compared to higher dimensional tasks.

R6 (other points): We did not use relabeled data to train the critic since we focus on task inference. The connection with
structural causal models is an interesting avenue for further work, but beyond scope of this submission. Our method is
not specific to BCQ. We will explain it more clearly in the final version. We will explain the tasks in the main text.
Finally, while our models are trained from MuJoCo states, they are high-dimensional. In HumanoidDir, the state has
376 dimensions. The task inference model input has 98560 dimensions.

(a) Our model outperforms baselines and ablations.

(b) Meta-GenRL's poor performance even on training task. Results obtain from official MetaGenRL code.

(c) Error on unseen task.

[Meta-Review · NeurIPS 2020]

Reviewers find the paper well-motivated and concisely written. While most of the techniques employed in the paper have been investigated in the literature, the work finds a bag of good tricks to solve the phenomenon the authors observed in multi-task batch RL where agents rely on shortcuts to identify tasks and hence do not generalize. Reviewers would like to see more expansion on related works, and better baselines and experiment environment to strengthen the work. Please try to incorporate these feedback when revising your draft.